# Effective Data Augmentation With Diffusion Models

**Brandon Trabucco** [1], **Kyle Doherty** [2], **Max Gurinas** [3], **Ruslan Salakhutdinov** [1]
[1] Carnegie Mellon University, [2] MPG Ranch, [3] University Of Chicago, Laboratory Schools
`brandon@btrabucco.com, rsalakhu@cs.cmu.edu`

## Abstract

Data augmentation is one of the most prevalent tools in deep learning, underpinning many recent advances, including those from classification, generative models, and representation learning. The standard approach to data augmentation combines simple transformations like rotations and flips to generate new images from existing ones. However, these new images lack diversity along key semantic axes present in the data. Current augmentations cannot alter the high-level semantic attributes, such as animal species present in a scene, to enhance the diversity of data. We address the lack of diversity in data augmentation with image-to-image transformations parameterized by pre-trained text-to-image diffusion models. Our method edits images to change their semantics using an off-the-shelf diffusion model, and generalizes to novel visual concepts from a few labelled examples. We evaluate our approach on few-shot image classification tasks, and on a real-world weed recognition task, and observe an improvement in accuracy in tested domains.

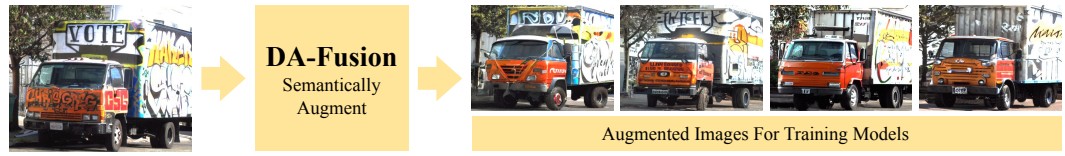

Figure 1: Real images (left) are semantically modified using a publicly available off-the-shelf Stable Diffusion checkpoint. Resulting synthetic images (right) are used for training downstream classification models.

## 1 Introduction

An omnipresent lesson in deep learning is the importance of internet-scale data, such as ImageNet (Deng et al., 2009), JFT (Sun et al., 2017), OpenImages (Kuznetsova et al., 2018), and LAION-5B (Schuhmann et al., 2022), which are driving advances in Foundation Models (Bommasani et al., 2021) for image generation. These models use large deep neural networks (Rombach et al., 2022) to synthesize photo-realistic images for a rich landscape of prompts. The advent of photo-realism in large generative models is driving interest in using synthetic images to augment visual recognition datasets (Azizi et al., 2023). These generative models promise to unlock diverse and large-scale image datasets from just a handful of real images without the usual labelling cost.

Standard data augmentations aim to diversify images by composing randomly parameterized image transformations (Antoniou et al., 2017; Perez and Wang, 2017; Shorten and Khoshgoftaar, 2019; Zhao et al., 2020). Transformations including flips and rotations are chosen that respect basic invariances present in the data, such as horizontal reflection symmetry for a coffee mug. Basic image transformations are thoroughly explored in the existing data augmentation literature, and produce models that are robust to color and geometry transformations. However, models for recognizing coffee mugs should also be sensitive to subtle details of visual appearance like the brand of mug; yet, basic transformations do not produce novel structural elements, textures, or changes in perspective. On the other hand, large pretrained generative models have become exceptionally sensitive to subtle visual details, able to generate uniquely designed mugs from a single example (Gal et al., 2022).

Our key insight is that large pretrained generative models *complement the weaknesses* of standard data augmentations, while *retaining the strengths*: universality, controllability, and performance. We propose a flexible data augmentation strategy that generates variations of real images using text-to-image diffusion models (DA-Fusion). Our method adapts the diffusion model to new domains by

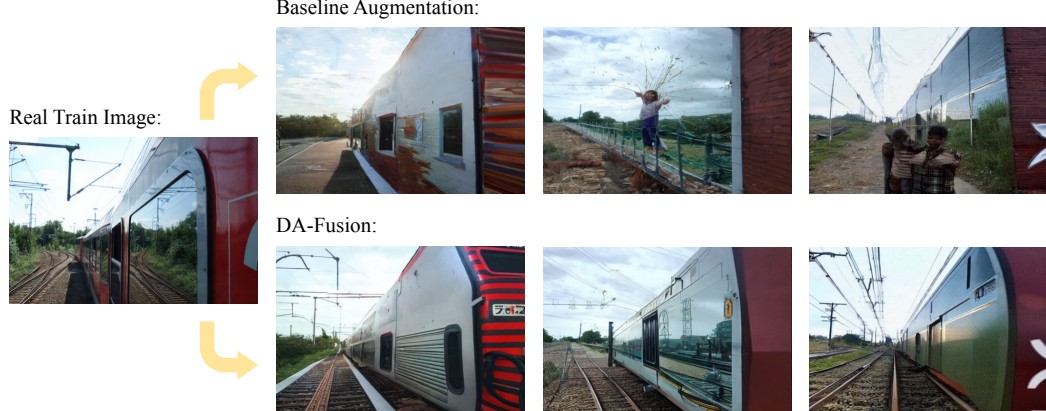

Figure 2: DA-Fusion produces task-relevant augmentations with no prior knowledge about the image content. Given an image of a train from PASCAL VOC (Everingham et al., 2009), we generate several augmentations using Real Guidance (He et al., 2022) (top row), and compare these to our method (bottom row).

fine-tuning pseudo-prompts in the text encoder representing concepts to augment. DA-Fusion modifies the appearance of objects in a manner that respects their semantic invariances, such as the design of the graffiti on the truck in Figure 1 and the design of the train in Figure 2. We test our method on few-shot image classification tasks with common and rare concepts, including a real-world weed recognition task the diffusion model has not seen before. Using the same hyper-parameters in all domains, our method outperforms prior work, improving data augmentation by up to +10 percentage points. Our ablations illustrate that DA-Fusion produces **larger gains for the more fine-grain concepts**. Open-source code is released at: `https://github.com/brandontrabucco/da-fusion`.

## 2 RELATED WORK

Generative models have been the subject of growing interest and rapid advancement. Earlier methods, including VAEs Kingma and Welling (2014) and GANs Goodfellow et al. (2014), showed initial promise generating realistic images, and were scaled up in terms of resolution and sample quality Brock et al. (2019); Razavi et al. (2019). Despite the power of these methods, many recent successes in photorealistic image generation were the result of diffusion models Ho et al. (2020); Nichol and Dhariwal (2021); Saharia et al. (2022b); Nichol et al. (2022); Ramesh et al. (2022). Diffusion models have been shown to generate higher-quality samples compared to their GAN counterparts Dhariwal and Nichol (2021), and developments like classifier free guidance Ho and Salimans (2022) have made text-to-image generation possible. Recent emphasis has been on training these models with internet-scale datasets like LAION-5B Schuhmann et al. (2022). Generative models trained at internet-scale Rombach et al. (2022); Saharia et al. (2022b); Nichol et al. (2022); Ramesh et al. (2022) have unlocked several application areas where photorealistic generation is crucial.

**Image Editing**    Diffusion models have popularized image-editing. Inpainting with diffusion is one such approach that allows the user to specify what to edit as a mask Saharia et al. (2022a); Lugmayr et al. (2022). Other works avoid masks and modify the attention weights of the diffusion process that generated the image instead Hertz et al. (2022); Mokady et al. (2022). Perhaps the most relevant technique to our work is SDEdit Meng et al. (2022), where real images are inserted partway through the reverse diffusion process. SDEdit is applied by He et al. (2022) to generate synthetic data for training classifiers, but our analysis differs from theirs in that we study generalization to new concepts the diffusion model wasn't trained on. To instruct the diffusion model on what to augment, we optimize a pseudo-prompt (Li and Liang, 2021; Gal et al., 2022) for each concept. Our strategy is more appealing than fine-tuning the whole model as in Azizi et al. (2023) since it works from just one example per concept (Azizi et al. (2023) require millions of images), and doesn't disturb the model's ability to generate other concepts. Our fine-tuning strategy improves the quality of augmentations for common concepts Stable Diffusion has seen, and for fine-grain concepts that are less common.

**Synthetic Data**    Training neural networks on synthetic data from generative models was popularized using GANs Antoniou et al. (2017); Tran et al. (2017); Zheng et al. (2017). Various applications for synthetic data generated from GANs have been studied, including representation learning Jahanian

et al. (2022), inverse graphics Zhang et al. (2021a), semantic segmentation Zhang et al. (2021b), and training classifiers Tanaka and Aranha (2019); Dat et al. (2019); Yamaguchi et al. (2020); Besnier et al. (2020); Xiong et al. (2020); Wickramaratne and Mahmud (2021); Haque (2021). More recently, synthetic data from diffusion models has also been studied in a few-shot setting He et al. (2022). These works use generative models that have likely seen images of target classes and, to the best of our knowledge, we present the first analysis for synthetic data on previously unseen concepts.

## 3 BACKGROUND

Diffusion models Sohl-Dickstein et al. (2015); Ho et al. (2020); Nichol and Dhariwal (2021); Song et al. (2021); Rombach et al. (2022) are sequential latent variable models inspired by thermodynamic diffusion Sohl-Dickstein et al. (2015). They generate samples via a Markov chain with learned Gaussian transitions starting from an initial noise distribution $p(x_T) = \mathcal{N}(x_T; 0, I)$.

$$p_\theta(x_{0:T}) = p(x_T) \prod_{t=1}^{T} p_\theta(x_{t-1}|x_t) \tag{1}$$

Transitions $p_\theta(x_{t-1}|x_t)$ are designed to gradually reduce variance according to a schedule $\beta_1, \ldots, \beta_T$ so the final sample $x_0$ represents a sample from the true distribution. Transitions are often parameterized by a fixed covariance $\Sigma_t = \beta_t I$ and a learned mean $\mu_\theta(x_t, t)$ defined below.

$$\mu_\theta(x_t, t) = \frac{1}{\sqrt{\alpha_t}} \left( x_t - \frac{\beta_t}{\sqrt{1 - \tilde{\alpha}_t}} \epsilon_\theta(x_t, t) \right) \tag{2}$$

This parameterization choice results from deriving the optimal reverse process Ho et al. (2020), where $\epsilon_\theta(\cdot)$ is a neural network trained to process a noisy sample $x_t$ and predict added noise. Given real samples $x_0$ and noise $\epsilon \sim \mathcal{N}(0, I)$, one can derive $x_t$ at an arbitrary timestep below.

$$x_t(x_0, \epsilon) = \sqrt{\tilde{\alpha}_t} x_0 + \sqrt{1 - \tilde{\alpha}_t} \epsilon \tag{3}$$

Ho et al. (2020) define $\alpha_t = 1 - \beta_t$ and $\tilde{\alpha}_t = \prod_{s=1}^{t} \alpha_t$. These components allow training and sampling from the type of diffusion model backbone in this work. We use a pretrained Stable Diffusion model trained by Rombach et al. (2022). Among other differences, this model includes a text encoder that enables text-to-image generation (refer to Appendix G for model details).

## 4 DATA AUGMENTATION WITH DIFFUSION MODELS

In this work we develop a flexible data augmentation strategy using text-to-image diffusion models. In doing so, we consider *three desiderata*: Our method is 1) **universal**: it produces high-fidelity augmentations for new and fine-grain concepts, not just the ones the diffusion model was trained on; 2) **controllable**: the content, extent, and randomness of the augmentation are simple to control and straightforward to tune; 3) **performant**: gains in accuracy justify the additional computational cost of generating images from Stable Diffusion. We discuss these in the following sections.

### 4.1 A UNIVERSAL GENERATIVE DATA AUGMENTATION

Standard data augmentations apply to all images regardless of class and content Perez and Wang (2017). We aim to capture this flexibility with our diffusion-based augmentation. This is challenging because real images may contain elements the diffusion model is not able to generate out-of-the-box. How do we generate plausible augmentations for such images? Shown in Figure 3, we adapt the diffusion model to new concepts by inserting $c$ new embeddings in the text encoder of the generative model, and fine-tuning only these embeddings to maximize the likelihood of generating new concepts.

**Adapting Generative Model** When generating synthetic images, previous work uses a prompt with the specified class name He et al. (2022). However, this is not possible for concepts that lie outside the vocabulary of the generative model because the model's text encoder has not learned words to describe these concepts. We discuss this problem in Section 5 with our contributed weed-recognition task, which our pretrained diffusion model is unable to generate when the class name is provided. A simple solution to this problem is to have the model's text encoder learn new words to describe new concepts. Textual Inversion (Gal et al., 2022) is well-suited for this, and we use it to learn a word embedding $\vec{w}_i$ from a handful of labelled images for each class in the dataset.

$$\min_{\vec{w}_0, \vec{w}_1, \ldots, \vec{w}_c} \mathbb{E} \left[ \|\epsilon - \epsilon_\theta(\sqrt{\tilde{\alpha}_t} x_0 + \sqrt{1 - \tilde{\alpha}_t} \epsilon, t, \text{"a photo of a } \vec{w}_i\text{"})\|^2 \right] \tag{4}$$

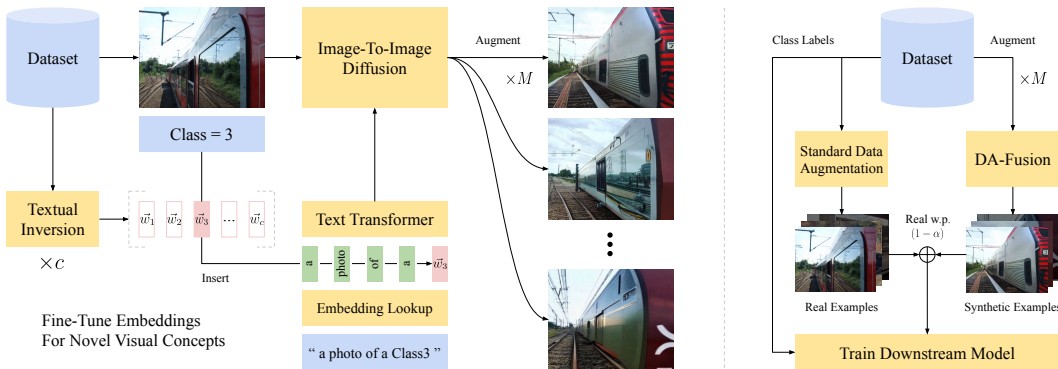

Figure 3: How our data augmentation works. Given a dataset of images and their class labels, we generate $M$ augmented versions of each real image using an image-editing technique and a pretrained Stable Diffusion checkpoint. Synthetic images are mixed with real data when training downstream models.

We initialize each new embedding $\vec{w}_i$ to a class-agnostic value (see Appendix G), and optimize them to minimize the simplified loss function proposed by Ho et al. (2020). Figure 3 shows how new embeddings $\vec{w}_i$ are inserted in the prompt given an image of a train. Our method is modular, and as other mechanisms are studied for adapting diffusion models, Textual Inversion can easily be swapped out with one of these, and the quality of the augmentations from DA-Fusion can be improved.

**Generating Synthetic Images** Many of the existing approaches generate synthetic images from scratch Antoniou et al. (2017); Tanaka and Aranha (2019); Besnier et al. (2020); Zhang et al. (2021b;a). This is particularly challenging for concepts the diffusion model hasn't seen before. Rather than generate from scratch, we use real images as a guide. We splice real images into the generation process of the diffusion model following prior work in SDEdit Meng et al. (2022). Given a reverse diffusion process with $S$ steps, we insert a real image $x_0^{\text{ref}}$ with noise $\epsilon \sim \mathcal{N}(0, I)$ at timestep $\lfloor St_0 \rfloor$, where $t_0 \in [0, 1]$ is a hyperparameter controlling the insertion position of the image.

$$x_{\lfloor St_0 \rfloor} = \sqrt{\tilde{\alpha}_{\lfloor St_0 \rfloor}} x_0^{\text{ref}} + \sqrt{1 - \tilde{\alpha}_{\lfloor St_0 \rfloor}} \epsilon \tag{5}$$

We proceed with reverse diffusion starting from the spliced image at timestep $\lfloor St_0 \rfloor$ and iterating Equation 2 until a sample is generated at timestep 0. Generation is guided with a prompt that includes the new embedding $\vec{w}_i$ for the class of the source image (see Appendix G for prompt details).

## 4.2 Controlling Augmentation

**Balancing Real & Synthetic Data** Training models on synthetic images often risks over-emphasizing spurious qualities and biases resulting from an imperfect generative model Antoniou et al. (2017). The common solution assigns different sampling probabilities to real and synthetic images to manage imbalance He et al. (2022). We adopt a similar method for balancing real and synthetic data in Equation 6, where $\alpha$ denotes the probability that a synthetic image is present at the $l$-th location in the minibatch of images $B$.

$$i \sim \mathcal{U}(\{1, \ldots, N\}), j \sim \mathcal{U}(\{1, \ldots, M\}) \tag{6}$$

$$B_{l+1} \leftarrow B_l \cup \left\{ X_i \text{ w.p. } (1-\alpha) \text{ else } \tilde{X}_{ij} \right\} \tag{7}$$

Here $X \in \mathcal{R}^{N \times H \times W \times 3}$ denotes a dataset of $N$ real images, and $i \in \mathbb{Z}$ specifies the index of a particular image $X_i$. For each image, we generate $M$ augmentations, resulting in a synthetic dataset $\tilde{X} \in \mathcal{R}^{N \times M \times H \times W \times 3}$ with $N \times M$ image augmentations, where $\tilde{X}_{ij} \in \mathcal{R}^{H \times W \times 3}$ enumerates the $j$th augmentation for the $i$th image in the dataset. Indices $i$ and $j$ are sampled uniformly from the available $N$ real images and their $M$ augmented versions respectively. Given indices $ij$, with probability $(1-\alpha)$ a real image image $X_i$ is added to the batch $B$, otherwise its augmented image $\tilde{X}_{ij}$ is added. Hyper-parameter details are presented in Appendix G, and we find $\alpha = 0.5$ to work effectively in all domains tested, which equally balances real and synthetic images.

**Improving Diversity By Randomizing Intensity** Having appropriately balanced real and synthetic images, our goal is to maximize diversity. This goal is shared with standard data augmentation Perez and Wang (2017); Shorten and Khoshgoftaar (2019), where multiple simple transformations are used, yielding more diverse data. Despite the importance of diversity, generative models typically employ frozen sampling hyperparameters to produce synthetic datasets Antoniou et al. (2017); Tanaka and Aranha (2019); Yamaguchi et al. (2020); Zhang et al. (2021b;a); He et al. (2022). Inspired by the success of randomization in standard data augmentations (such as the angle of rotation), we propose to randomly sample the insertion position $t_0$ where real images are spliced into Equation 5. This can be interpreted as randomizing the extent images are modified—as $t_0 \rightarrow 0$ generations more closely resemble the guide image.

In Section 6.2 we sample uniformly at random $t_0 \sim \mathcal{U}(\{\frac{1}{k}, \frac{2}{k}, \ldots, \frac{k}{k}\})$, and observe a consistent improvement in classification accuracy with $k = 4$ compared to fixing $t_0$. Though the hyperparameter $t_0$ is perhaps the most direct translation of randomized intensity to generative model-based data augmentations, there are several alternatives. For example, one may consider the guidance scale parameter used in classifier-free guidance (Ho and Salimans, 2022). We leave this as future work.

## 5 DATA PREPARATION

**Standard Datasets** We benchmark our data augmentations on six standard computer vision datasets. We employ Caltech101 (Fei-Fei et al., 2004), Flowers102 (Nilsback and Zisserman, 2008), FGVC Aircraft (Maji et al., 2013), Stanford Cars (Krause et al., 2013), COCO Lin et al. (2014), and PASCAL VOC Everingham et al. (2009). We use the official 2017 training and validation sets of COCO, and the official 2012 training and validation sets of PASCAL VOC. We adapt these datasets into object classification tasks by filtering images that have at least one object segmentation mask. We assign these images labels corresponding to the class of object with largest area in the image, as measured by the pixels contained in the mask. Caltech101, COCO, and PASCAL VOC have *common* concepts like "dog" and Flowers102, FGVC Aircraft, and Stanford Cars have *fine-grain* concepts like "giant white arum lily" (the specific flower name). Additional details for preparing datasets are in Appendix G.

**Leafy Spurge** We contribute a dataset of top-down drone images of semi-natural areas in the western United States. These data were gathered in an effort to better map the extent of a problematic invasive plant, leafy spurge (*Euphorbia esula*), that is a detriment to natural and agricultural ecosystems in temperate regions of North America. Prior work to classify aerial imagery of leafy spurge achieved an accuracy of 0.75 Yang et al. (2020). To our knowledge, top-down aerial imagery of leafy spurge was not present in the Stable Diffusion training data. Results of

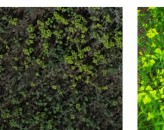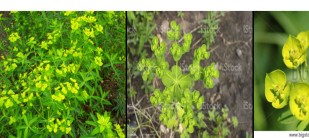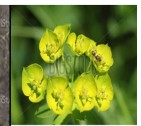

Figure 4: A sample from the Spurge dataset (the first on the left), compared with top results of CLIP-retrieval queried on the prompt: "a drone image of leafy spurge". Closeup images from members of the same genus (second, and third) are in the top 20 results and a closeup of the same species for the 35th result (fourth). No top-down aerial images of the target plant were revealed.

CLIP-retrieval Beaumont (2022) returned close-up, side-on images of members of the same genus (Figure 4) in the top 20 results. We observed the first instance of our target species, *Euphorbia esula*, as a 35th result. The spurge images we contribute are semantically distinct from those in the CLIP corpus, because they capture the plant and landscape context around it from 50m distance above the ground, rather than close-up botanical features. Therefore, this dataset represents a unique opportunity to explore few-shot learning with Stable Diffusion, and developing a robust classifier would directly benefit efforts to restore natural ecosystems. Additional details are in Appendix H.

## 6 DA-FUSION IMPROVES FEW-SHOT CLASSIFICATION

**Experimental Details** We test few-shot classification on seven datasets with three data augmentation strategies. RandAugment (Cubuk et al., 2020) employs no synthetic images, and uses the default hyperparameters in `torchvision`. Real Guidance (He et al., 2022) uses SDEdit on real images with $t_0 = 0.5$, has a descriptive prompt about the class, and shares hyperparameters with our method to ensure fair evaluation. DA-Fusion is prompted with "a photo of a $<w_i>$" where the embedding for $<w_i>$ is initialized to the embedding of the class name and learned according to Section 4.1.

Each real image is augmented $M$ times, and a ResNet50 classifier pre-trained on ImageNet is fine-tuned on a mixture of real and synthetic images sampled as discussed in Section 4.2. We vary the

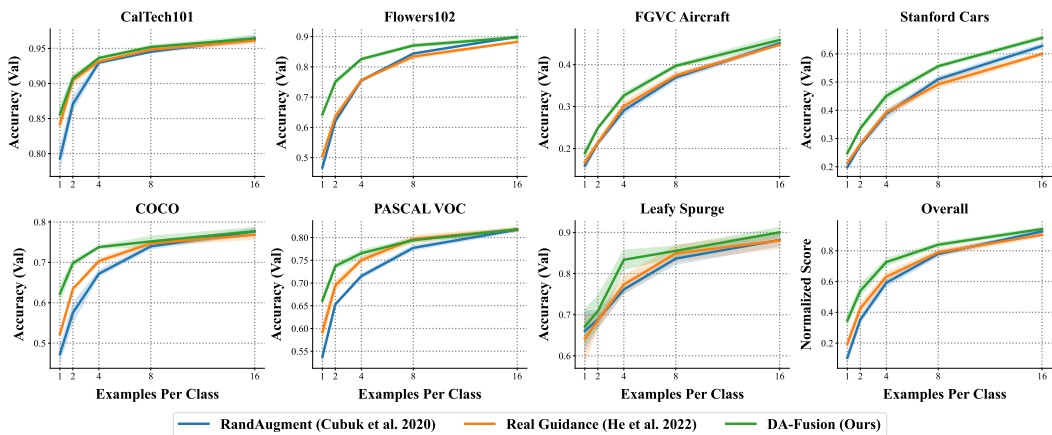

Figure 5: Few-shot classification performance with full information. DA-Fusion consistently outperforms RandAugment (Cubuk et al., 2020), and Real Guidance (He et al., 2022) with a descriptive prompt. In fine-grain domains such as Flowers102, which represents classification of flowers into subclasses like "giant white arum lily," Real Guidance performs no better than traditional data augmentation. In contrast, DA-Fusion performs consistently well across a variety of domains with common concepts (COCO, PASCAL VOC, Caltech101), rare concepts (Flowers102, FGVC Aircraft, Stanford Cars) and novel concepts to Stable Diffusion (Leafy Spurge).

number of examples per class used for training the classifier on the x-axis in the following plots, and fine-tune the final linear layer of the classifier for $10,000$ steps with a batch size of 32 and the Adam optimizer with learning rate $0.0001$. We record validation metrics every 200 steps and report the epoch with highest accuracy. Solid lines in plots represent means, and error bars denote 68% confidence intervals over 4 independent trials. An overall score is calculated for all datasets after normalizing performance using $y_i^{(d)} \leftarrow (y_i^{(d)} - y_{\min}^{(d)})/(y_{\max}^{(d)} - y_{\min}^{(d)})$, where $d$ represents the dataset, $y_{\max}^{(d)}$ is the maximum performance for any trial of any method, and $y_{\min}^{(d)}$ is defined similarly.

**Interpreting Results**   Results in Figure 5 show DA-Fusion improves accuracy in every domain, often by a significant margin when there are few real images per class. We observe gains between +5 and +15 accuracy points in all seven domains compared to standard data augmentation. Our results show how generative data augmentation can significantly outperform color and geometry-based transformations like those in RandAugment (Cubuk et al., 2020). Despite using a powerful generative model with a descriptive prompt, Real Guidance He et al. (2022) performs inconsistently, and in several domains fails to beat RandAugment. To understand this behavior, we binned the results by whether a dataset contains common concepts (COCO, PASCAL VOC, Caltech101), fine-grain concepts (Flowers102, FGVC Aircraft, Stanford Cars), or completely new concepts (Leafy Spurge), and visualized the normalized scores for the three data augmentation methods in Figure 6.

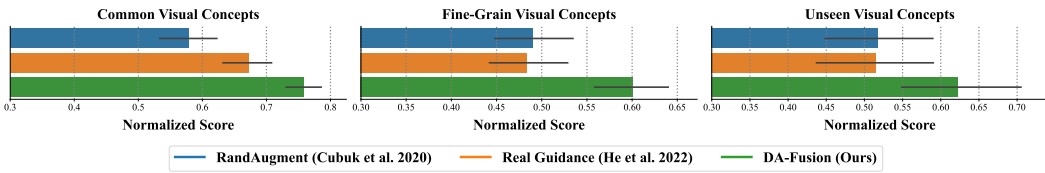

Figure 6: Performance stratified by concept novelty. Real Guidance (He et al., 2022) uses a descriptive prompt to instruct Stable Diffusion what to augment, and works for common concepts Stable Diffusion was trained on. However, for harder-to-describe concepts, this strategy fails. DA-Fusion works well at all novelty levels, improving by 12.8% for common concepts, 24.2% for fine-grain concepts, and 20.8% for unseen concepts.

**Class Novelty Hinders Real Guidance**   Figure 6 reveals a systematic failure mode in Real Guidance (He et al., 2022) for novel and fine-grain concepts. These concepts are harder to describe in a prompt than common ones—consider the prompts "a top-down drone image of leafy spurge taken from 100ft in the air above a grassy field" versus "a photo of a cat." DA-Fusion mitigates this by optimizing pseudo-prompts, formatted as "a photo of a $<w_i>$", that instruct the diffusion model on what to generate, and has the added benefit of requiring no prompt engineering. Our method works well at all

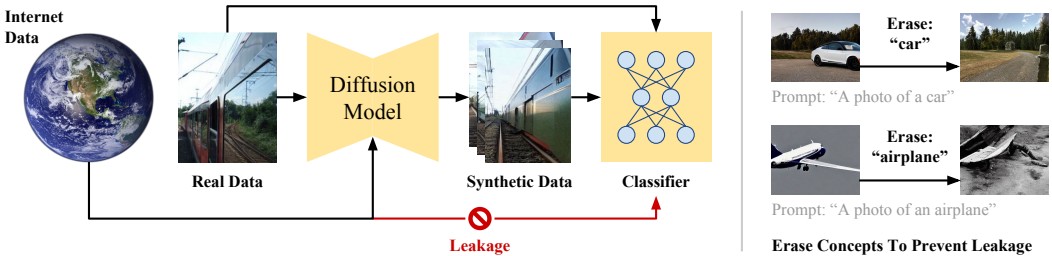

Figure 7: Leakage of internet data to downstream models. Large generative models trained at internet scale may produce synthetic data similar to their training data when tested on common concepts (right). We show the result of erasing common concepts by fine-tuning the attention layer weights of Stable Diffusion's UNet (right).

levels of concept novelty, and produces larger gains the more fine-grain concepts are, improving by 12.8% for common concepts, 24.2% for fine-grain concepts, and 20.8% for novel concepts.

## 6.1 PREVENTING LEAKAGE OF INTERNET DATA

Previous work utilizing large pretrained generative models to produce synthetic data (He et al., 2022) has left an important question unanswered: *are we sure they are working for the right reason?* Models trained on internet data have likely seen many examples of classes in common benchmarking datasets like ImageNet Deng et al. (2009). Moreover, Carlini et al. (2023) have recently shown that pretrained diffusion models can leak their training data. Leakage of internet data, as in Figure 7, risks compromising evaluation. Suppose our goal is to test how images from diffusion models improve few-shot classification with only a few real images, but leakage of internet data gives our classifier access to thousands of real images. Performance gains observed may not reflect the quality of the data augmentation methodology itself, and may lead to drawing the wrong conclusions.

We explore two methods for preventing leakage of Stable Diffusion's training data. We first consider a *model-centric* approach that prevents leakage by editing the model weights to remove class knowledge. We also consider a *data-centric* approach that hides class information from the model inputs.

**Model-Centric Leakage Prevention**   Our goal with this approach is to remove knowledge about concepts in our benchmarking datasets from the weights of Stable Diffusion. We accomplish this by fine-tuning Stable Diffusion in order to remove the ability to generate concepts from our benchmarking datasets. Given a list of class names in these datasets, we utilize a recent method developed by Gandikota et al. (2023) that fine-tunes the UNet backbone of Stable Diffusion so that concepts specified by a given prompt can no longer be generated (we use class names as such prompts). In particular, the UNet is fine-tuned to minimize the following loss function.

$$\min_{\theta} \mathbb{E}\left[\left\|\epsilon_\theta(x_t, t, \text{"class name"}) - \epsilon_{\theta^*}(x_t, t) + \eta(\epsilon_{\theta^*}(x_t, t, \text{"class name"}) - \epsilon_{\theta^*}(x_t, t))\right\|^2\right] \quad (8)$$

Where "class name" is replaced with the actual class name of the concept being erased, $\theta$ represents the parameters of the UNet being fine-tuned, and $\theta^*$ represents the initial parameters of the UNet. This procedure, named ESD by Gandikota et al. (2023), can be interpreted as guiding generation in the opposite direction of classifier free-guidance, and can erase a variety of types of concepts.

**Data-Centric Leakage Prevention**   While editing the model directly to remove knowledge about classes is a strong defense against possible leakage, it is also costly. In our experiments, erasing a single class from Stable Diffusion takes two hours on a single 32GB V100 GPU. As an alternative for situations where the cost of a model-centric defense is too high, we can achieve a weaker defense by removing all mentions of the class name from the inputs of the model. In practice, switching from a prompt that has the class name to a new prompt omitting the class name is sufficient. Section 4.1 goes into detail how to implement this defense for different types of models.

**Results With Model-Centric Leakage Prevention**   Figure 8 shows results when erasing class knowledge from Stable Diffusion weights. We observe a consistent improvement in validation accuracy by as much as +5 percentage points on the Pascal and COCO domains when compared to the standard data augmentation baseline. DA-Fusion exceeds performance of Real Guidance He

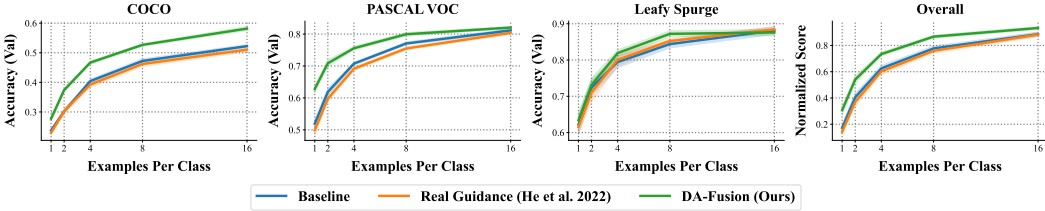

Figure 8: Few-shot classification performance with model-centric leakage prevention. We evaluate DA-Fusion on three classification datasets and outperform classic data augmentation and a competitive method from recent literature. We report validation accuracy as a function of the number of images in the observed training set in the top row, and the average accuracy over the interval (the number of examples per class) in the bottom row.

Figure 9: Few-shot classification performance with data-centric leakage prevention. DA-Fusion unilaterally outperforms the standard data augmentation baseline, and a competitive method from recent literature. Gains with DA-Fusion are larger with this weaker defense against training data leakage compared to the stronger model-centric defense. We discuss how to interpret this larger improvement in Section 6.

et al. (2022) overall while utilizing the same hyperparameters, without any prior information about the classes in these datasets. In this setting, Real Guidance performs comparably to the baseline, which suggests that gains in Real Guidance may stem from information provided by the class name. This experiment shows DA-Fusion improves few-shot learning and suggests our method generalizes to concepts Stable Diffusion wasn't trained on. To understand how these gains translate to weaker defenses against training data leakage, we next evaluate our method using a data-centric strategy.

**Results With Data-Centric Leakage Prevention**  Figure 9 shows results when class information is hidden from Stable Diffusion inputs. As before, we observe a consistent improvement in validation accuracy, by as much as +10 percentage points on the Pascal and COCO domains when compared to the standard data augmentation baseline. DA-Fusion exceeds performance of Real Guidance He et al. (2022) in all domains while utilizing the same hyperparameters, without specifying the class name as an input to the model. With a weaker defense against training data leakage, we observe larger gains with DA-Fusion. This suggests gains are due in part to accessing Stable Diffusion's prior knowledge about classes, and highlights the need for a strong leakage prevention mechanism when evaluating synthetic data from large generative models. We next ablate our method to understand where these gains come from, and how important each part of the method is.

### 6.2   HOW IMPORTANT ARE RANDOMIZED INTENSITIES?

Our goal in this section is to understand what fraction of gains are due to randomizing the intensity of our augmentation based on Section 4.2. We employ the same experimental settings as in Section 6, using data-centric leakage prevention, and run our method using a fixed insertion position $t_0 = 0.5$ (labelled $k = 1$ in Figure 10), following the settings used with Real Guidance. In Figure 10 we report the improvement in average classification accuracy on the validation set versus standard data augmentation. These results show that both versions of our method outperform the baseline, and randomization improves our method in all domains, leading to an overall improvement of 51%.

### 6.3   DA-FUSION IS ROBUST TO DATA BALANCE

We next conduct an ablation to understand the sensitivity of our method to the balance of real and synthetic data, controlled by two hyperparameters: the number of synthetic images per real image $M \in \mathbb{N}$, and the probability of sampling synthetic images during training $\alpha \in [0, 1]$. We use $\alpha = 0.5$ and $M = 10$ throughout the paper. Insensitivity to the particular value of $\alpha$ and $M$ is a desireable

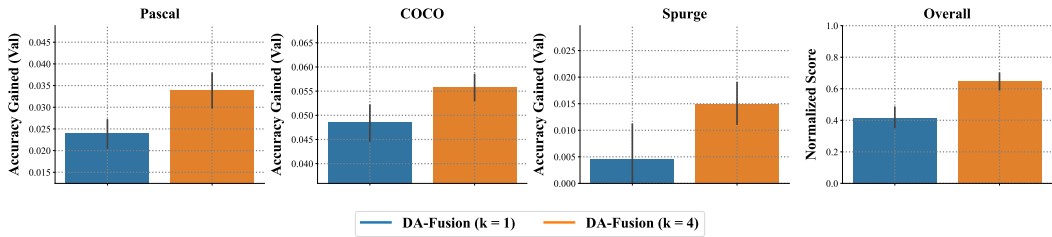

Figure 10: Ablation for randomizing augmentation intensity. We vary the number of intensities from ($k = 4$) in the main experiments, to a deterministic insertion position $t_0 = 0.5$. We report the improvement in average few-shot classification accuracy over standard data augmentation, and observe a consistent improvement.

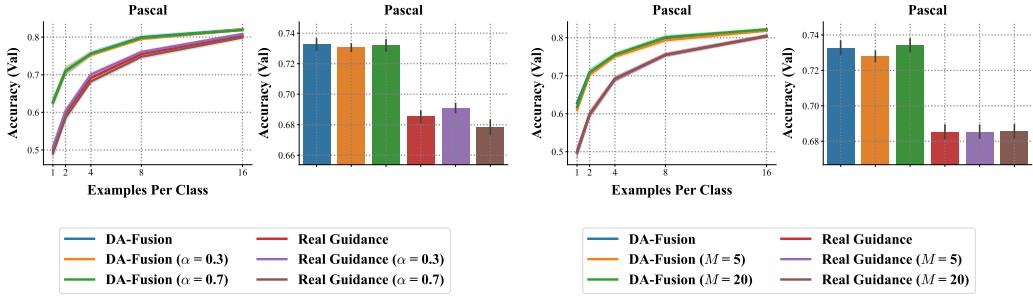

Figure 11: Ablation for data balance sensitivity. We run our method with $\alpha \in \{0.3, 0.5, 0.7\}$ and $M \in \{5, 10, 20\}$ and report the improvement in few-shot classification accuracy over Real Guidance using the same settings. DA-Fusion is robust to the balance of real and synthetic data and outperforms prior work in each setting.

trait because it simplifies hyper-parameter tuning and facilitates our data augmentation working out-of-the-box with no domain-specific tuning. We test sensitivity to $\alpha$ and $M$ by comparing runs of DA-Fusion with different assignments to Real Guidance with the same $\alpha$ and $M$. Figure 11 shows stability as $\alpha$ and $M$ varies, and that $\alpha = 0.7$ performs marginally better than $\alpha = 0.5$, which suggests our method improves synthetic image quality because sampling them more often improves accuracy. While $M = 20$ performs marginally better than $M = 10$, the added cost of doubling the number of generative model calls for a marginal improvement suggests $M = 10$ is sufficient.

## 7 DISCUSSION

We proposed a flexible method for data augmentation based on diffusion models, DA-Fusion. Our method adapts a pretrained diffusion model to semantically modify images and produces high quality augmentations regardless of image content. Our method improves few-shot classification accuracy in tested domains, and by up to +10 percentage points on various datasets. Similarly, our method produces gains on a contributed weed-recognition dataset that lies outside the vocabulary of the diffusion model. To understand these gains, we studied how performance is impacted by potential leakage of Stable Diffusion training data. To prevent leakage during evaluation, we presented two defenses that target the model and data respectively, each on different sides of a trade-off between defense strength and computational cost. When subject to both defenses, DA-Fusion consistently improves few-shot classification accuracy, which highlights its utility for data augmentation.

There are several directions to improve the flexibility and performance of our method as future work. First, our method does not explicitly control *how* an image is augmented by the diffusion model. Extending the method with a mechanism to better control how objects in an image are modified, e.g. changing the breed of a cat, could improve the results. Recent work in prompt-based image editing Hertz et al. (2022) suggests diffusion models can make localized edits without pixel-level supervision, and would minimally increase the human effort required to use DA-Fusion. This extension would let image attributes be handled independently by DA-Fusion, and certain attributes could be modified more extremely than others. Second, data augmentation is becoming increasingly important in the decision-making setting Yarats et al. (2022). Maintaining temporal consistency is an important challenge faced when using our method in this setting. Solving this challenge could improve the few-shot generalization of policies in complex visual environments. Finally, improvements to our diffusion model backbone that enhance image photo-realism are likely to improve DA-Fusion.

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

## A    LIMITATIONS & SAFEGUARDS

As generative models have improved in terms of fidelity and scale, they have been shown to occasionally produce harmful content, including images that reinforce stereotypes, and images that include nudity or violence. Synthetic data from generative models, when it suffers from these problems, has the potential to increase bias in downstream classifiers trained on such images if not handled. We employ two mitigation techniques to lower the risk of leakage of harmful content into our data augmentation strategy. First, we use a safety checker that determines whether augmented images contain nudity or violence. If they do, the generation is discarded and re-sampled until a clean image is returned. Second, rather than generate images from scratch, our method edits real images, and keeps the original high-level structure of the real images. In this way, we can guide the model away from harmful content by ensuring the real images contain no harmful content to begin with. The combination of these techniques lowers the risk of leakage of harmful content, but is not a perfect solution. In particular, detecting biased content that encourages racial or gender stereotypes that exist online is much harder than detecting nudity or violence, and one limitation of this work is that we can't yet defend against this. We emphasize the importance of curating unbiased and safe datasets for training large generative models, and the creation of post-training bias mitigation techniques.

## B    ETHICAL CONSIDERATIONS

There are potential ethical concerns arising from large-scale generative models. For example, these models have been trained on large amounts of user data from the internet without the explicit consent of these users. Since our data augmentation strategy employs Stable Diffusion (Rombach et al., 2022), our method has the potential to generate augmentations that resemble or even copy data from such users online. This issue is not specific to our work; rather, it is inherent to image generation models trained at scales as large as Stable Diffusion, and other works using Stable Diffusion also face this ethical problem. Our mitigation to this ethical problem is to allow deletion of concepts from the weights of Stable Diffusion before augmentation. Deletion removes harmful, or copyrighted material from Stable Diffusion weights to ensure it cannot be copied by the model during augmentation.

## C    BROADER IMPACTS

Data augmentation strategies like DA-Fusion have the potential to enable training vision models of a variety of types from limited data. While we studied classification in this work, DA-Fusion may also be applied to video classification, object detection, and visual reinforcement learning. One risk associated with improved few-shot learning on vision-based tasks is that synthetic data can be generated targeting particular users. For example, suppose one intends to build a person-identification system used to record the behavior patterns of a specific person in public. Such a system

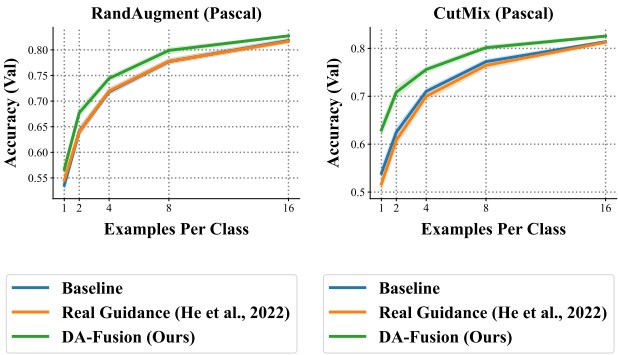

Figure 12: Results with stronger data-augmentation baselines. DA-Fusion improves over both RandAugment (using default PyTorch settings), and CutMix (with default setting from Yun et al. (2019)). Note that RandAugment results use model-centric leakage prevention, and CutMix results use data-centric leakage prevention, showing we improve over stronger baselines in both regimes.

trained with generative model-based data augmentations may only need one real photo to be trained. This poses a risk to privacy, despite other benefits that few-shot learning provides. As another example, suppose one intends to build a system capable of generating pornography of a specific celebrity. Few-shot learning makes this possible with just a handful of real images that exist online. This poses a risk to personal safety and bodily autonomy of the targeted person.

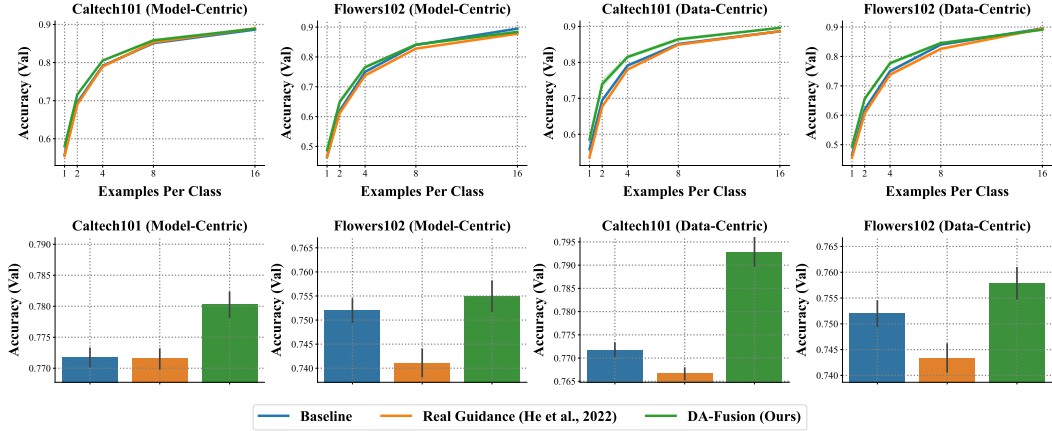

Figure 13: Few-shot classification performance with both kinds of leakage prevention on additional datasets. DA-Fusion outperforms the standard data augmentation baseline, and a competitive method from recent literature. These results reinforce the message in the main paper: DA-Fusion is an effective data augmentation strategy.

## D  ADDITIONAL RESULTS

We conduct additional experiments on the Caltech101 (Fei-Fei et al., 2004), and Flowers102 (Nilsback and Zisserman, 2008) datasets, two standard image classification tasks. These tasks are commonly used when benchmarking few-shot classification performance, such as in the Visual Task Adaptation Benchmark (Zhai et al., 2019). Results on these datasets are shown in Figure 13, and show that DA-Fusion improves classification performance both when using a model-centric defense against training data leakage, and a data-centric defense, described in Section 6.1 of the paper.

## E  STRONGER AUGMENTATION BASELINES

In the main paper, we considered data augmentation baselines consisting only of randomized rotations and flips. In this section, we compare against two stronger data augmentation methods: RandAugment (Cubuk et al., 2020), and CutMix (Yun et al., 2019). Results are presented in Figure 12, and show that DA-Fusion improves over both RandAugment and CutMix on the Pascal-based task.

## F  DIFFERENT CLASSIFIER ARCHITECTURES

Results in the main paper use a ResNet50 architecture for the image classifier. In this section, we consider the Data-Efficient Image Transformer (DeiT) (Touvron et al., 2021), and evaluate DA-Fusion with data-centric leakage prevention on the Pascal task. Results in Figure 14 show that DA-Fusion improves the performance of DeiT, and suggests that gains generalize to different model architectures, including both convolution-based models (such as ResNet50), and attention-based ones (such as ViT).

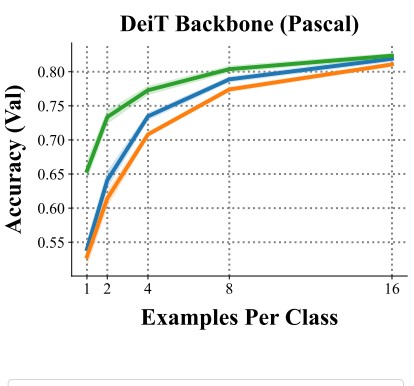

Figure 14: Few-shot results with a stronger classification model. DA-Fusion improves DeiT when compared to standard data augmentation baseline, and Real Guidance.

## G  HYPERPARAMETERS

Our method inherits the hyperparameters of text-to-image diffusion models and SDEdit Meng et al. (2022). In addition, we introduce several other hyperparameters in this work that control the diversity of the synthetic images. Specific values for these hyperparameters are given in Table 1.

| Hyperparameter Name | Value |
| --- | ---: |
| Synthetic Probability $\alpha$ | 0.5 |
| Real Guidance Strength $t_0$ | 0.5 |
| Num Intensities $k$ | 4 |
| Intensities Distribution $t_0$ | $\mathcal{U}(\{0.25, 0.5, 0.75, 1.0\})$ |
| Synthetic Images Per Real $M$ | 10 |
| Synthetic Images Per Real $M$ (spurge) | 50 |
| Textual Inversion Token Initialization | "the" |
| Textual Inversion Batch Size | 4 |
| Textual Inversion Learning Rate | 0.0005 |
| Textual Inversion Training Steps | 1000 |
| Class Agnostic Prompt | "a photo" |
| Standard Prompt | "a photo of a <class name>" |
| Textual Inversion Prompt | "a photo of a ClassX" |
| Stable Diffusion Checkpoint | `CompVis/stable-diffusion-v1-4` |
| Stable Diffusion Guidance Scale | 7.5 |
| Stable Diffusion Resolution | 512 |
| Stable Diffusion Denoising Steps | 1000 |
| Classifier Architecture | ResNet50 |
| Classifier Learning Rate | 0.0001 |
| Classifier Batch Size | 32 |
| Classifier Training Steps | 10000 |
| Classifier Early Stopping Interval | 200 |

Table 1: Hyperparameters and their values.

We uniformly at random select 20 classes per dataset for evaluation, turning them into 20-way classification tasks. This reduces the computational cost of reproducing the results in our paper, and the exact classes used in each dataset can be found in the open-source code.

## H   Leafy Spurge Dataset Acquisition and Pre-processing

In June 2022 botanists visited areas in western Montana, United States known to harbor leafy spurge and verified the presence or absence of the target plant at 39 sites. We selected sites that represented a range of elevation and solar input values as influenced by terrain. These environmental axes strongly drive variation in the structure and composition of vegetation Amatulli et al. (2018); Doherty et al. (2021). Thus, stratifying by these aspects of the environment allowed us to test the performance of classifiers when presented with a diversity of plants which could be confused with our target.

During surveys, each site was divided into a 3 x 3 grid of plots that were 10m on side (**Fig. 15**), and then botanists confirmed the presence or absence of leafy spurge within each grid cell. After surveying we flew a DJI Phantom 4 Pro at 50m above the center of each site and gathered still RGB images. All images were gathered on the same day in the afternoon with sunny lighting conditions.

We then cropped the the raw images to match the bounds of plots using visual markers installed during surveys as guides (**Fig. 16**). Resulting crops varied in size because of the complexity of terrain. E.G., ridges were closer to the drone sensor than valleys. Thus, image side lengths ranged from 533 to 1059 pixels. The mean side length was 717 and the mean spatial resolution, or ground sampling distance, of pixels was 1.4 cm.

In our initial hyperparameter search we found that the classification accuracy of plot-scale images was less than that of a classifier trained on smaller crops of the plots. Therefore, we generated four 250x250 pixel crops sharing a corner at plot centers for further experimentation (**Fig. 17**). Because spurge plants were patchily distributed within a plot, a botanist reviewed each crop in the present class and removed cases in which cropping resulted in samples where target plants were not visually apparent.

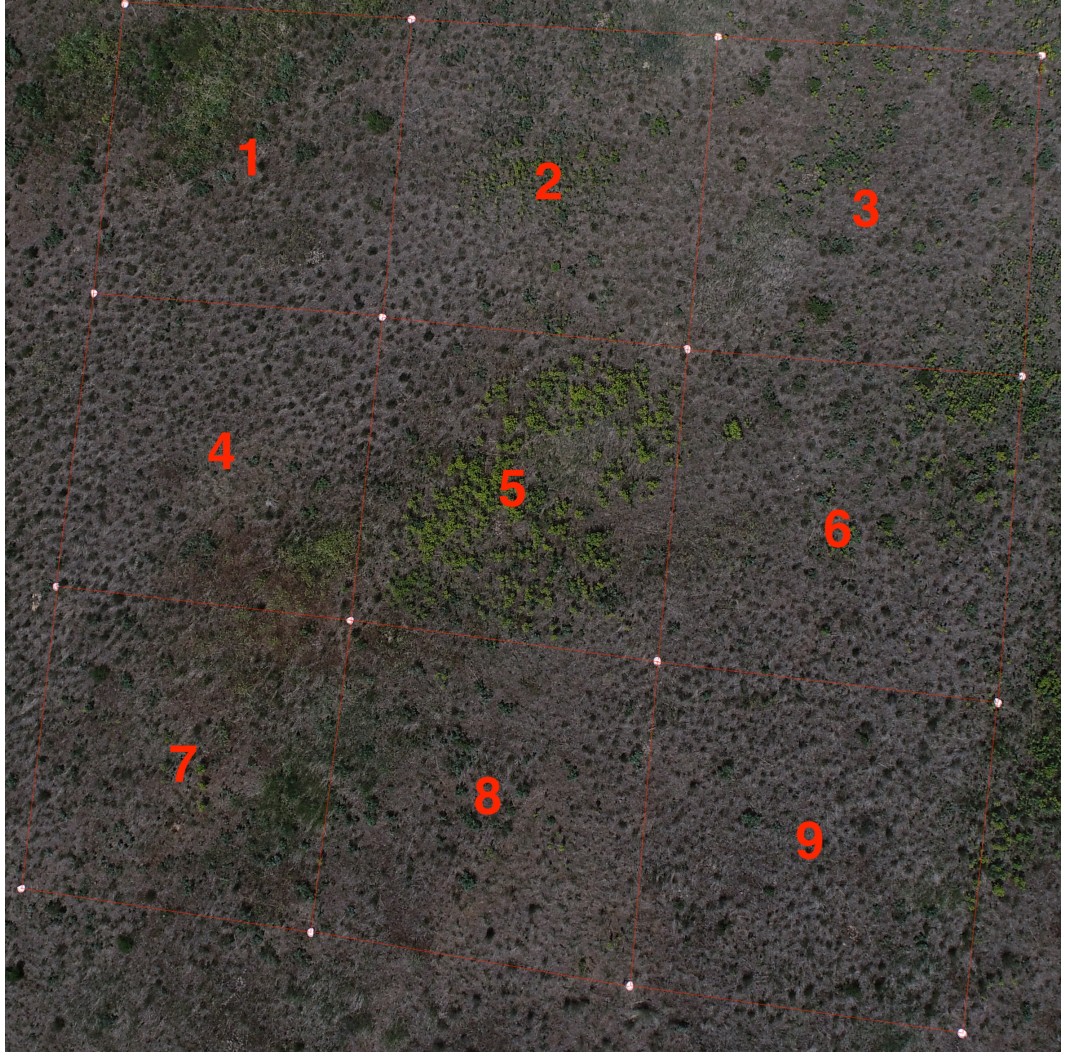

Figure 15: A drone image of surveyed areas containing leafy spurge. At each site botanists verified spurge presence or absence in a grid of nine spatially distinct plots. Note that cell five is rich in leafy spurge.

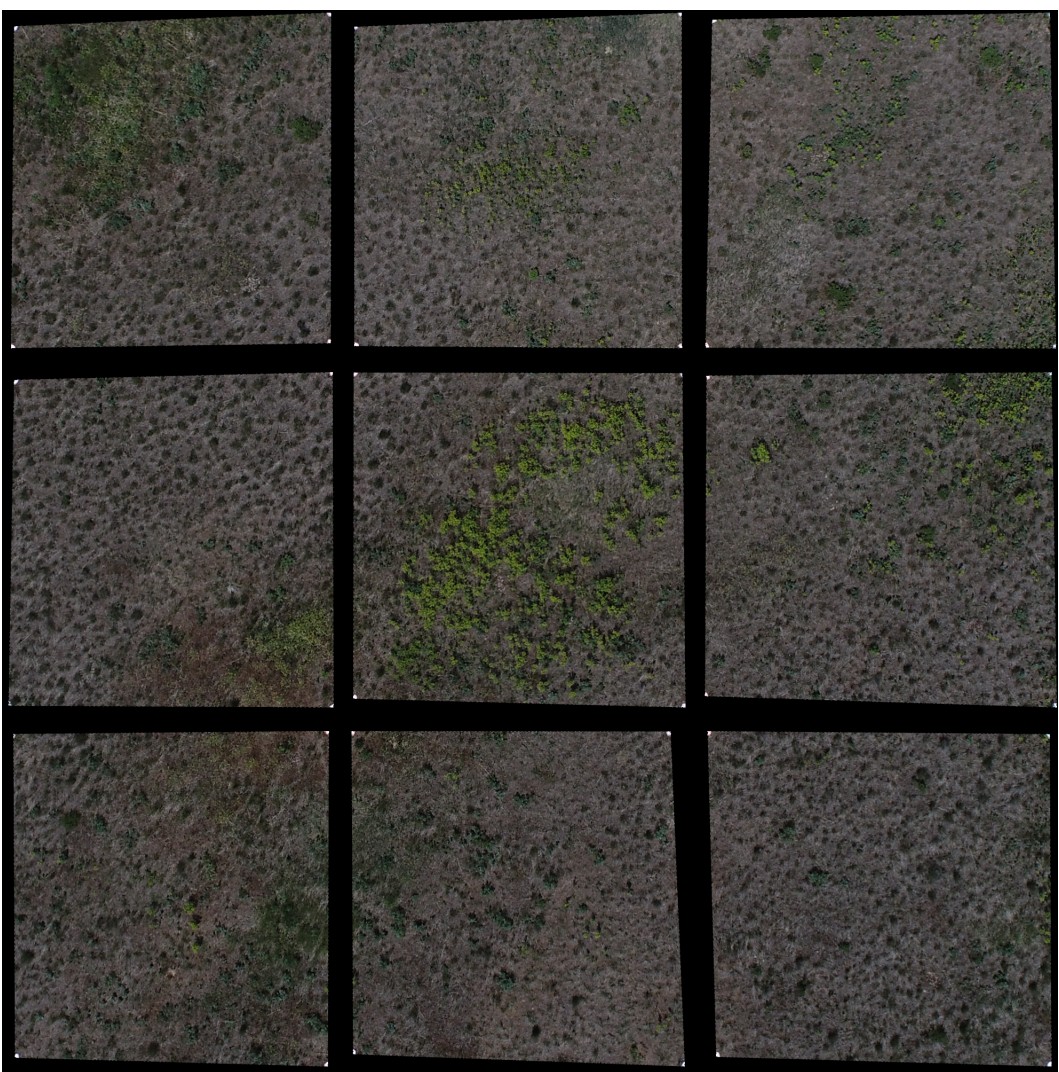

Figure 16: Markers installed at the corners of plots were used to crop plots from source images.

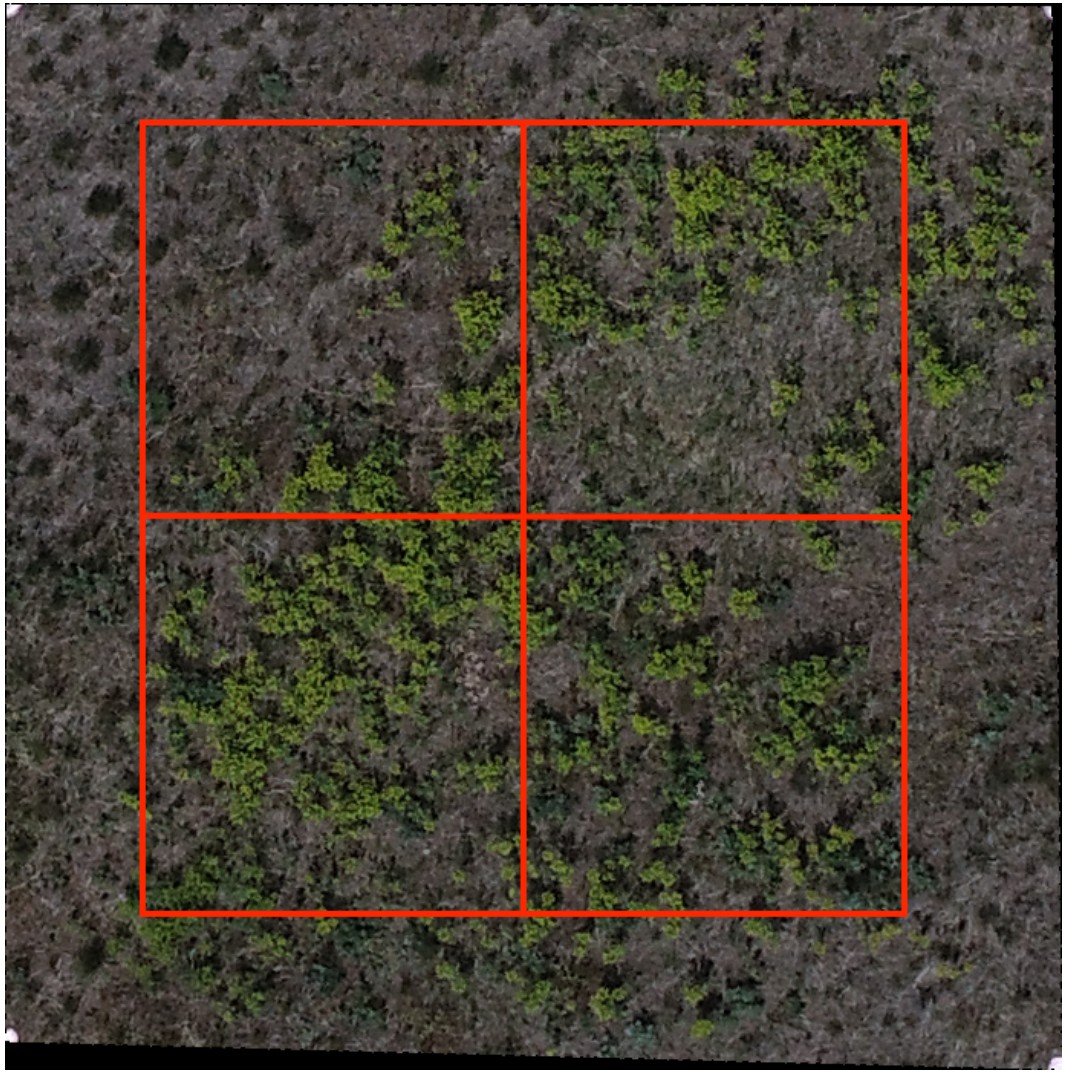

Figure 17: At each plot image center we cropped four 250x250 pixel sub-plots. We did this to amplify our data and improve classifier performance. The crops of plots with spurge present labels were inspected by a botanist to filter out examples where cropping excluded the target plant or the plants were not apparent.

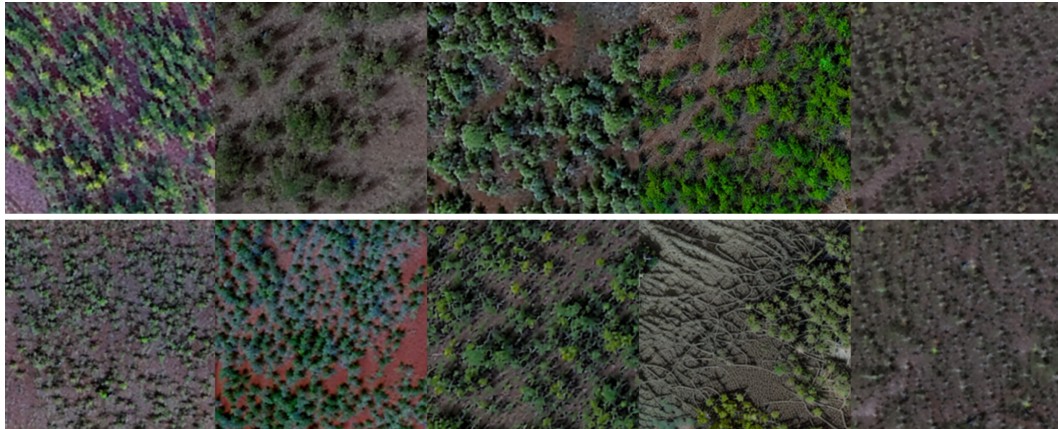

Figure 18: Here we show examples of synthetic images generated from the leafy spurge dataset with DA-Fusion methods. The top row shows output where images are pooled to fine-tune a single token per class. The bottom row shows examples where tokens are generated specifically for each image. Source images, inference hyperparameters, and seed are otherwise identical in each column.

## I   BENCHMARKING THE LEAFY SPURGE DATASET

We benchmark classifier performance here on the full leafy spurge dataset, comparing a baseline approach incorporating legacy augmentations with our novel DA-fusion method. For 15 trials we generated random validation sets with 20 percent of the data, and fine-tuned a pretrained ResNet50 on the remaining 80 percent using the training hyperparameters reported in section **??** for 500 epochs. From these trials we compute cross-validated mean accuracy and 68 percent confidence intervals.

In the case of baseline experiments, we augment data by flipping vertically and horizontally, as well as randomly rotating by as much as 45 degrees with a probability of 0.5. For DA-Fusion augmentations we take two approaches(**Fig. 18**) The first we refer to as DA-Fusion Pooled, and we apply the methods of Textual Inversion Gal et al. (2022), but include all instances of a class in a single session of fine-tuning, generating one token per class. In the second approach we refer to as DA-Fusion Specific, we fine-tune and generate unique tokens for each image in the training set. In the specific case, we generated 90, 180, and 270 rotations as well as horizontal and vertical flips and contribute these along with original image for Stable Diffusion fine-tuning to achieve the target number of images suggested to maximize performanceGal et al. (2022). In both DA-Fusion approaches we generated ten synthetic images per real image for model training. We maintain $\alpha = 0.5$, evenly mixing real and synthetic data during training. We also maximize synthetic diversity by randomly selecting 0.25, 0.5, 0.75, and 1.0 $t_0$ values. Note that we do not apply concept erasure here as in few-shot experiments from the body text.

Both approaches to DA-Fusion offer slight performance enhancements over baseline augmentation methods for the full leafy spurge dataset. We observe a 1.0% gain when applying DA-Fusion Pooled and a 1.2% gain when applying DA-Fusion Specific(**Fig. 19**). It is important to note that, as implemented currently, compute time for DA-Fusion Specific is linearly related to data amount, but DA-Fusion Pooled compute is the same regardless of data size.

While pooling was not the most beneficial in this experiment, we support investigating it further. This is because fine-tuning a leafy spurge token in a pooled approach might help to orient our target in the embedding space where plants with similar diagnostic properties, such as flower shape and color from the same genus, may be well represented. However, the leafy-spurge negative cases do not correspond to a single semantic concept, but a plurality, such as green fields, brown fields, and wooded areas. It is unclear if fine-tuning a single token for negative cases by a pooled method would remove diversity from synthetic samples of spurge-free background landscapes, relative to an image-specific approach. For this reason, we suspect a hybrid approach of pooled token for the positive case and specific tokens for the negative cases could offer further gains, and support the application of detecting weed invasions into new areas.

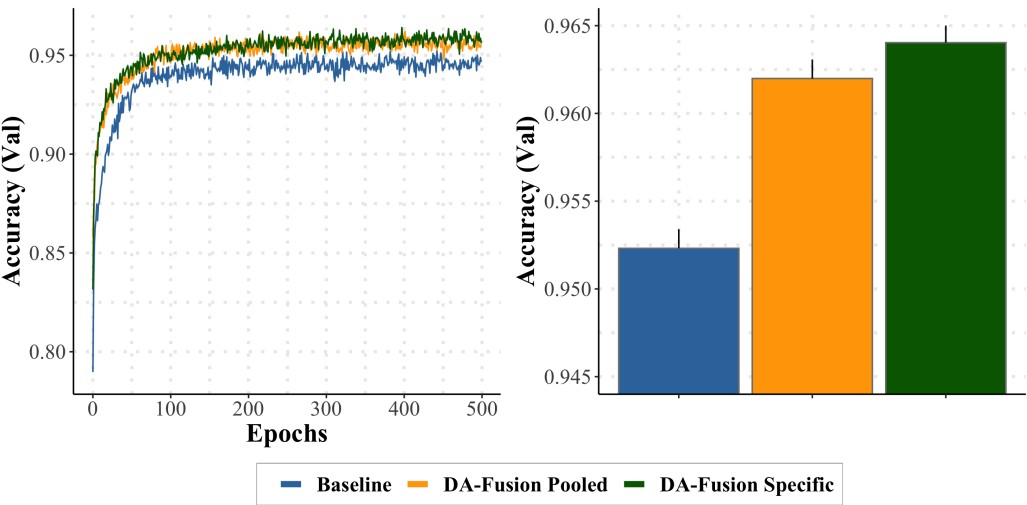

Figure 19: Cross-validated accuracy of leafy spurge classifiers when trained with baseline augmentations versus DA-Fusion methods on the full dataset. In addition to the benefits of DA-Fusion in few-shot contexts, we also find our method improves performance on larger datasets. Generating image-specific tokens (green line and bar) offers the most gains over baseline, though at the cost of greater compute.

