# OpenReview forum: "Effective Data Augmentation With Diffusion Models"
_ICLR.cc/2024/Conference — ICLR 2024 poster_

### Official Review · Reviewer_5pTF · 2023-10-19

**Soundness:** 3 good
**Presentation:** 4 excellent
**Contribution:** 2 fair
**Rating:** 6
**Confidence:** 4

**Summary:**

The authors proposed DA-Fusion, which uses an image-to-image diffusion model to generate new synthetic images to assist classification tasks. DA-Fusion utilizes Textual Inversion to learn the word embeddings of unseen concepts and employs the data balancing and random intensity trick to improve the augmentation results further. The empirical shows the effectiveness of DA-Fusion in the low-shot setting on common concepts, fine-grained concepts, and rare concepts.

**Strengths:**

- The paper shows positive empirical results on several classification tasks covering common, fine-grained, and rare concepts.

- The paper is well-written and easy to follow.

**Weaknesses:**

(1) The technical contribution is limited. DA-Fusion combines several existing methods, like the image-to-image diffusion model, the Textual Inversion, and the data balancing technique.

(2) DA-Fusion outperforms the much simpler RandAugment method mostly on extremely low-shot settings (less than 16 images per class). The improvements of DA-Fusion seem to be marginal when there are more than 16 images per class. Also, it is unfair to compare with RandAugment, which only uses the default hyperparameters, while DA-Fusion is “fine-tuned” on the target data (i.e., textual inversion, selecting $M$, and other parameters). The authors should search for the optimal number of operations and augmentation magnitude for RandAugment for a fairer comparison.

(3) The authors tested their method on COCO and PASCAl VOC datasets for pure classification but not object detection. Being a general image-to-image diffusion model, the generation step in DA-Fusion may alter the position of the objects in input images. It seems that DA-Fusion can only be applied to classification tasks.

**Questions:**

- The authors used a different number of augmented images per real image, $M$ ($M$=50 for spurge and $M$=10 for other data). Are there any guidelines for selecting $M$?

- As pointed out in the weakness, the technical contribution of this work is slightly limited. It is recommended to compare DA-Fusion with more and stronger data augmentation baselines (instead of just RandAugment and Real Guidance) to claim a more significant empirical contribution.

---

> ### Author Response · Authors · 2023-11-18
> **Response To Reviewer 5pTF (1/2)**
>
> Thank you for reviewing our paper and for your feedback. The points discussed in this review center around the strength of the technical contribution, the fairness of the evaluation, the selection of hyperparameters, and the applicability of the method beyond classification tasks. We address these and provide clarifications in the following rebuttal.
>
> # DA-Fusion’s Technical Contribution
>
> __“The technical contribution is limited. DA-Fusion combines several existing methods, like the image-to-image diffusion model, the Textual Inversion, and the data balancing technique”__
>
> DA-Fusion is a performant method for augmenting classification datasets. On seven tasks, we outperform RandAugment (Cubuk et al. 2019, https://arxiv.org/abs/1909.13719), a popular data augmentation technique, and Real Guidance (He et al., 2023, https://arxiv.org/abs/2210.07574), a state-of-the-art method for generative data augmentation presented at ICLR 2023. Our subsequent analysis in Figure 6 highlights a failure mode of the state-of-the-art method: it performs no better than RandAugment for fine-grain and unseen visual concepts. Addressing this failure mode is crucial for generative data augmentation so it performs well for all kinds of images, not just images with common concepts Stable Diffusion has seen in its training data.
>
> We are contributing the first generative data-augmentation technique with strong performance on fine-grain and unseen visual concepts (see Figure 6 in the paper). The simplicity of the method can be viewed as a strength: it does not require specialized components for strong performance. Simplicity has the advantage of (1) fewer hyperparameters, (2) straightforward reproducibility, and (3) modularity. These are desirable traits for a data-augmentation technique.

---

> ### Author Response · Authors · 2023-11-18
> **Response To Reviewer 5pTF (2/2)**
>
> # Evaluation & Hyperparameter Selection
>
> __“it is unfair to compare with RandAugment, which only uses the default hyperparameters, while DA-Fusion is fine-tuned”__
>
> DA-Fusion and RandAugment use a fixed set of hyperparameters across all tasks in the paper (excluding Leafy Spurge, which we are updating in the manuscript per the additional experiment noted in a later response). We did not fine-tune the hyperparameters of DA-Fusion per task, and doing so for either method would risk biasing the evaluation. The hyperparameters of RandAugment were shown in Cubuk et al. 2019 (https://arxiv.org/abs/1909.13719) to work well across several standard datasets.
>
> __“The authors should search for the optimal number of operations and augmentation magnitude for RandAugment for a fairer comparison.”__
>
> This would risk unfairly favoring RandAugment in our evaluation. We use a uniform set of hyperparameters with DA-Fusion and do the same with RandAugment and Real Guidance. Note that standard data augmentation can be applied in parallel with DA-Fusion (see Figure 3 in the paper), and improvements in performance due to fine-tuning RandAugment per task would also improve the performance of DA-Fusion by a corresponding amount.
>
> __“Are there any guidelines for selecting M?”__
>
> We are happy to include a discussion of the role and selection of M in the next revision of the manuscript. Introduced in Section 4.2, this hyperparameter controls the number of synthetic images generated per real image before training starts. The larger M, the higher the computational cost of DA-Fusion, but the higher performance can be attained. We ablate M in Figure 11 and find that our method is not sensitive to its value, observing that larger values of M perform marginally better than smaller values. M should be chosen based on the computational constraints of the user, but we find M=10 is an effective value across standard datasets.
>
> __“compare DA-Fusion with more and stronger data augmentation baselines (instead of just RandAugment and Real Guidance)”__
>
> At time of submission, the most recent and performant generative data augmentation is Real Guidance, adapted from He et al. 2023 (https://arxiv.org/abs/2210.07574). Being the state-of-the-art method, Real Guidance is the strongest and most relevant baseline to include. RandAugment is chosen for inclusion due to being recent, popular, and performant. We include an additional comparison against CutMix (Figure 12), another recent, popular, and performant data augmentation, and results are consistent with the existing comparison to RandAugment.
>
> If you have any specific additional data augmentation baselines to compare to DA-Fusion, we are happy to include them in the next revision of the manuscript.
>
> __“The authors used a different number of augmented images per real image, M (M=50 for spurge and M=10 for other data).”__
>
> We have evaluated DA-Fusion on the new Leafy Spurge task with the same value of M as the rest of the datasets in Figure 5 and will add this in the manuscript. The new experiment is consistent with the existing results in the paper: that DA-Fusion continues to outperform RandAugment and Real Guidance. Results can be viewed at this anonymous link:
> https://drive.google.com/drive/folders/1cSPeNTOmZK-vTTrxF5xvtEtBtzrYRMXd?usp=sharing
>
> # Applicability For Object Detection
>
> __“DA-Fusion may alter the position of the objects in input images … It seems that DA-Fusion can only be applied to classification tasks.”__
>
> Our focus in this work is classification, but our public code supports masks for objects, allowing DA-Fusion to independently transform objects in the image, and the background. (This is the relevant section)[https://github.com/anonymous-da-fusion/da-fusion/blob/master/semantic_aug/augmentations/textual_inversion.py#L152 ] of the anonymous code. By using masks, DA-Fusion can preserve the location of objects in the image. Examples using DA-Fusion with masks to preserve the locations of objects are in the following anonymous google drive link. This functionality allows our method to be applied to datasets that require object locations to be preserved during augmentation, such as object detection and segmentation tasks. We are adding a section discussing this application of DA-Fusion in the appendix of the manuscript.
> https://drive.google.com/drive/folders/15O0HWkQsEskmzmI8F14ISf2xNdwOlKBG?usp=sharing

---

> ### Author Response · Authors · 2023-11-21
> **Following Up On The Rebuttal**
>
> Thank you for reviewing our manuscript, we would like to follow up on our rebuttal.  If there are any remaining points that you would like us to address, please let us know.  Otherwise, thank you for your review and we look forward to your response.

---

> ### Comment · Reviewer_5pTF · 2023-11-21
>
> Thanks for the response. The authors addressed my concerns about the technical contribution and experiment settings. It is good that DA-Fusion can be applied to object detection tasks. Based on the additional studies, I am increasing my score from 5 to 6.

---

### Official Review · Reviewer_hrLX · 2023-10-28

**Soundness:** 3 good
**Presentation:** 3 good
**Contribution:** 3 good
**Rating:** 8
**Confidence:** 4

**Summary:**

This paper propose a diffusion model-based data augmentation technique for image classification. The method is based on a pretrained diffusion model. Textual inversion is applied to learn word embeddings for the classes in the target dataset for which the pretrained diffusion model may not learn before. During the generative process, real image is inserted at some time step to guide the generation. When training the target model, real image and synthetic image are mixed by probabilistic sampling. The paper further discuss some design choices for the proposed method, including the time step at which the real image is inserted during generation, strategies to prevent leakage of internet data, mixing ratio of real and synthetic images and the number of augmentations generated for each image. The proposed method is benchmarked on seven classification datasets and shown to outperform RandAugment and Real Guidance.

**Strengths:**

1. The proposed method that combines several existing techniques for data augmentation is interesting.
2. The paper provide insightful analysis on several design choices, including the time step at which the real image is inserted during generation, strategies to prevent leakage of internet data, mixing ratio of real and synthetic images and the number of augmentations generated for each image.
3. The paper will release code and an aerial imagery dataset of leafy spurge, which will facilitate future study in this direction.

**Weaknesses:**

1. The proposed technique is only applicable for classification tasks. It is not clear how it can be applied for object detection and segmentation tasks. My thinking is that one of the major drawbacks of such generative model-based augmentation method vs. traditional method may be that it can not simultaneously generate the segmentation mask and bounding box annotation for the augmented images.

**Questions:**

1. I am not clear how the data-centric leakage prevention is performed. The paper mention that "switching from a prompt ... is sufficient" and "Section 4.1 goes into detail". If data-centric leakage refers to using prompt like "a photo of a Class3", what is the setting difference between Fig.5 and Fig.9?

---

> ### Author Response · Authors · 2023-11-20
> **Response To Reviewer hrLX**
>
> Thank you for reviewing our paper, and for your valuable feedback! We appreciate your questions and positive impression of the paper. In this rebuttal, we hope to answer your questions, and discuss the points in your review to address the weakness you listed. The main weakness discussed in this review is the applicability of DA-Fusion beyond classification tasks to object detection and semantic segmentation. We agree with the reviewer that data augmentation has been crucial for object detection and semantic segmentation models, and we discuss in this rebuttal how DA-Fusion can be applied out-of-the-box to such tasks.
>
> # Detection & Segmentation
>
> __“one of the major drawbacks of such generative model-based augmentation method vs. traditional [data augmentation] may be that it can not simultaneously generate the segmentation mask and bounding box annotation for the augmented images”__
>
> Applying DA-Fusion to images with segmentation mask and bounding box annotations requires either (1) generating the annotations with the image, or (2) taking existing annotations and augmenting the corresponding input images in a manner that preserves the annotations. Our anonymous public code __supports capability (2) out-of-the-box__, and (here is the relevant section)[https://github.com/anonymous-da-fusion/da-fusion/blob/master/semantic_aug/augmentations/textual_inversion.py#L152] of the anonymous code. By using masks, DA-Fusion can preserve the location of objects in the input images. Examples using DA-Fusion with masks to preserve the locations of objects are in the following anonymous google drive link. We are adding a section discussing this application of DA-Fusion in the appendix of the manuscript.
> https://drive.google.com/drive/folders/15O0HWkQsEskmzmI8F14ISf2xNdwOlKBG?usp=sharing
>
> # Answers To Questions
>
> __“If data-centric leakage refers to using prompt like ‘a photo of a Class3’, what is the setting difference between Fig.5 and Fig.9?”__
>
> Data-centric leakage prevention modifies the prompt used with Stable Diffusion to omit the name of the class, and we discuss the technical details in the paragraph titled “Data-Centric Leakage Prevention” in Section 6.1 of the manuscript. The high-level difference between Figure 5 and Figure 9 is whether Stable Diffusion knows the name of the class of the image. If the class name is known as in Figure 5, we prompt Real Guidance with “a photo of a <class-name>” and we initialize the learned pseudo-prompt for DA-Fusion with the class name.
>
> If we don’t know the class name as in Figure 9, we prompt Real Guidance with “a photo” and we initialize the learned pseudo-prompt for DA-Fusion with the word “the”. These values are discussed in Table 1 of Appendix G - Hyperparameters. The ablation in Figure 9 shows that DA-Fusion works well for concepts that Stable Diffusion may not have learned the name of.

---

### Official Review · Reviewer_KxYv · 2023-10-31

**Soundness:** 2 fair
**Presentation:** 3 good
**Contribution:** 2 fair
**Rating:** 6
**Confidence:** 4

**Summary:**

The paper introduces DA-Fusion that generates additional training data using Stable Diffusion. It involves learning a word embedding that represents each class in the dataset and using SDEdit for generation. Experimental results are shown for few-shot classification on different types of datasets e.g., common concepts, rare concepts, etc, with supportive results.

**Strengths:**

- The paper was well written and easy to follow.
- Performing textual inversion for novel concepts seems like a promising idea.

**Weaknesses:**

1. Generalizability of the method.
    - It seems like the learned tokens are helpful for learning dataset specific biases, as it can add additional information on the general “style” of the dataset and of its classes. However, it seems to make certain assumptions about the datasets, e.g., that the images are object centric - only the target class is present, they have standard poses/viewpoints. It is not clear if there is substantial variations within the images of that class, e.g. in iwildcam, where the images from each class can be from different camera traps, thus, different backgrounds, camera parameters, the animals can have very different poses, or can be highly occluded. In this case, what would the single word embedding learn?

2. Desideratas (sec 4)
    - Controllable (content)
        - Is textual inversion (as it is used in the paper) controllable? I.e. Is it easy to control for whether the token learns the style or object? Or if certain objects tends to co-occur with another, e.g. train and rails, it may not be necessarily learning the token for trains but also the objects that co-occur with it.
    - “performant: gains in accuracy justify the additional computational cost of generating images from Stable Diffusion”
        - There does not seem to be any comparisons with compute cost relative to the baseline Rand Aug. DA-Fusion requires performing textual inversion, which may be expensive.
        - It would also be interesting to compare the performance with standard augs like Rand Aug for different M. E.g., [1] showed that training on synthetic data outperforms when the generated dataset size much larger than the original dataset size. It is possible that at smaller M, standard augmentations outperform synthetic data, with the additional benefit that it is also fast to compute.

3. Experiments
    - Randaug, which includes several color based augmentations does not seem to be a good baseline for fine grained datasets like Flowers102, that may require the original color for classification.


[1] Sariyildiz et al. Fake it till you make it: Learning transferable representations from synthetic ImageNet clones. CVPR’23

**Questions:**

In addition to the questions in weakness, some clarifications about the experiments:
- How does the method perform zero shot? If the learned tokens did capture the class characteristics, and there is no distribution shift between training and test, it seems like DA-Fusion should do well.
- Is the leafy spurge task a binary classification problem i.e., detecting if there is leafy spurge in the image or not? If that is the case, how often do other plants occur in the data? I was wondering how much fine-grained details can textual inversion capture.
- How much variance is there in the class word embedding? If textual inversion is performed again with a different seed, how different would the generations be? If they are, it can be another way to introduce diversity.

---

> ### Author Response · Authors · 2023-11-17
> **Response To Reviewer KxYv (1 / 3)**
>
> Thank you for reviewing our paper and for your feedback. The discussion points shared by this reviewer center around (1) the generalizability of the method to challenging datasets, (2) controlling what visual concepts our method learns when images have co-occurring objects, and (3) the cost of generating synthetic data compared to standard data augmentations.
>
> We address these points in our rebuttal.
>
> # Generalizing To Challenging Datasets:
>
>
> __“it seems to make certain assumptions about the datasets, e.g., that the images are object centric - only the target class is present, they have standard poses/viewpoints”__
>
>
> Generative data augmentations should generalize to a variety of challenging datasets, including those with atypical viewpoints, poses, and objects. We test this in our paper by adapting the COCO and PASCAL VOC object detection tasks into classification problems. While the remaining standard datasets we use for evaluation tend to make certain assumptions that simplify benchmarking, COCO and PASCAL VOC have images with __many co-occurring objects__ in a __variety of poses__, with __occlusions__, and in __atypical viewpoints__. DA-Fusion outperforms prior work on these tasks by up to +10 percentage points.
>
> Here are images from the COCO dataset used in our evaluation that satisfy the properties discussed by this reviewer (the following links are anonymous):
>
> (__first image__): https://farm4.staticflickr.com/3664/3300381750_6bca02bce3_z.jpg
>
> http://farm1.staticflickr.com/89/263427144_db4970197f_z.jpg
>
> http://farm9.staticflickr.com/8102/8453554475_6a130a1c3b_z.jpg
>
> The first image linked above includes (1) a close-up view of a fire-hydrant, (2) multiple people in a variety of poses, (3) one of whom is occluded by the fire-hydrant, (4) and three partially-occluded cars. DA-Fusion has strong performance on COCO and PASCAL VOC tasks (see Figure 5 in the paper), despite these challenging properties of the images.
>
>
> __“different backgrounds, camera parameters, the animals can have very different poses, or can be highly occluded. In this case, what would the single word embedding learn?”__
>
>
> Textual Inversion is a powerful technique for learning visual concepts. Gal et al. 2022 (https://arxiv.org/abs/2208.01618) show Textual Inversions for objects with different backgrounds, viewpoints, poses, and partial occlusions. See Figures 3-5 in Gal et al. 2022. Textual Inversion is effective in these conditions and generates objects and styles well.
>
> The embeddings found by Textual Inversion capture a distribution. They represent an object in a variety of poses, with a variety of occlusions, and on a variety of backgrounds.

---

> ### Author Response · Authors · 2023-11-17
> **Response To Reviewer KxYv (2 / 3)**
>
> # Controlling What DA-Fusion Generates:
>
> __“Is it easy to control for whether the token learns the style or object? … it may not be necessarily learning the token for trains but also the objects that co-occur with it.”__
>
> Generative models are uniquely promising for data augmentation because they allow the user to control the content and style of images in training datasets. DA-Fusion implements three simple interfaces that control the content and style of what is generated by our method.
>
> 1) guiding what’s generated with prompts
>
> 2) guiding what Textual Inversion learns with prompts
>
> 3) erasing the ability to generate specific concepts written in prompts
>
> Each control method has a simple natural language interface.
>
> __Control Method 1__: The word embeddings learned by DA-Fusion are flexible, and the user can modify the content and style of generations after learning the word embeddings. For example, if the prompt “a photo of a <class>” leads to the images named “original generation” at the following anonymous link, changing the prompt to “a photo of a <class> wearing a bow-tie”, leads to the images named “generation with bow-tie” at the following anonymous link:
> https://drive.google.com/drive/folders/1gnCa0SBd9tadL9o2SK0jmNNYoNvC76Sc?usp=sharing
> We are updating the manuscript to discuss __Control Method 1__ in the appendix.
>
> __Control Method 2__: Suppose we show DA-Fusion the image named “original image” at the following anonymous link and learn a “<class>” word embedding using this image. The image has a cat and a dog. If we want the “<class>” word embedding to focus on the cat and ignore the dog, we can prompt Stable Diffusion with “a photo of a <class> sitting on the right of a dog” when learning the “<class>” word embedding. If we generate images using an inference-time prompt “a photo of a <class>” the generations only have cats, showing the “<class>” word embedding has ignored the dog and focused on the cat in the original image.
> https://drive.google.com/drive/folders/1lmr7e7qBZHwyGJyIkxt-0bUAxcn0934M?usp=sharing
> We are updating the manuscript to discuss __Control Method 2__ in the appendix.
>
> Controlling the style of what tokens learn can be done with Textual Inversion. We build on the Textual Inversion script released by HuggingFace, and by using `--learnable_property=style` the user can switch modes and cause tokens to learn style rather than object-based concepts.
>
> __Control Method 3__: Finally, we can erase Stable Diffusion’s generation ability for specific concepts. Suppose we generate images of roads using the prompt “a photo of a <class>” and we realize that cars often show up in the generations, but our real data never has cars. We can erase the “car” concept and prevent Stable Diffusion from generating it, resulting in the following images. Additional examples for erasing concepts are in Figure 7 of the manuscript.
> https://drive.google.com/drive/folders/1eDkXFKhpN94ZEhQtih97PdXsJtjzqKuz?usp=sharing
> We discuss __Control Method 3__ in Section 6.1 of the manuscript.
>
> # Computational Cost Analysis:
>
> __“There does not seem to be any comparisons with compute cost relative to the baseline Rand Aug. DA-Fusion requires performing textual inversion, which may be expensive.”__
>
> The trade-off of performance and computational cost is important for users considering the viability of generative data augmentations. We agree the cost of DA-Fusion is higher than traditional data augmentation, but our method is parallelizable, and the cost is once upfront, whereas the cost of traditional data augmentation is at every training iteration.
>
> We provide statistics for DA-Fusion in the following table, assuming a dataset with 100 classes, 4 images per class, a training batch size of 32, trained for 10,000 steps, with M=5. The machine has 4 Nvidia RTX 6000 ada GPUs, and image generation is parallelized across all the GPUs.
>
> | Method | Total Time (minutes) |
> | ----------- | ----------- |
> | DA-Fusion (25-step diffusion sampler) | 21:19 |
> | DA-Fusion (12-step diffusion sampler) | 14:47 |
> | RandAugment | 12:18 |
> | No Augmentation | 08:15 |
>
> DA-Fusion is only 20.2% more expensive than RandAugment with the 12-step diffusion sampler according to wall-clock time in this configuration, and performs up to 24.2% better for fine-grain concepts, which is discussed in the paper in Section 6, Figure 6 in the manuscript. We are adding this cost analysis to the paper.

---

> > ### Author Response · Authors · 2023-11-17
> > **Response To Reviewer KxYv (3 / 3)**
> >
> > # Answers To Questions:
> >
> > __“How does the method perform zero shot?”__
> >
> > Requiring a handful of real images is an intuitive interface for users of DA-Fusion because they can show our method what kind of images they want to generate. Being fully zero-shot, the user would have to describe all aspects of the generated images in the prompt, which can be difficult. While DA-Fusion could be adapted to this setting, we found it works best when real images are used as a guide (see the paragraph titled “Generating Synthetic Images” in Section 4.1). Previous works already show synthetic data from Diffusion Models improves classification when no real data is available, such as in Azizi et al. 2023, https://arxiv.org/abs/2304.08466.
> >
> > __“Is the leafy spurge task a binary classification problem i.e., detecting if there is leafy spurge in the image or not? If that is the case, how often do other plants occur in the data? I was wondering how much fine-grained details can textual inversion capture.”__
> >
> > The leafy spurge task is binary: the target is present or absent. Other plant species are always present in the leafy spurge images. The composition of plant species was highly varied and capturing this diversity was intentional on the part of the authors, who hoped to represent the breadth of plant communities. Many spurge plants are immediately adjacent to other plants, and often partially obscured by them. For example, a shrub or tree may contribute overstory and obscure leafy spurge plants. Additionally, the authors captured spurge at various stages in the plant’s life history, where some individuals in an image show distinct yellow flowers, while others do not. Leafy spurge plants may cluster spatially as their colonies expand, forming large contiguous areas. The degree to which plants are clustered is also varied throughout the dataset and even in individual images. You might encounter an individual in one region of the image, and a continuous mat of individuals in another region of the same image.
> >
> > __“How much variance is there in the class word embedding?”__
> >
> > Different seeds of Textual Inversion lead to different image generations. We have uploaded generations for different Textual Inversion seeds for cats from the PASCAL VOC task at the following anonymous link. This could provide additional diversity for DA-Fusion.
> > https://drive.google.com/drive/folders/16uOtwsks53mTnXe2DEaDCLFNhQ_zc-yJ?usp=sharing
> >
> > __“It is possible that at smaller M, standard augmentations outperform synthetic data, with the additional benefit that it is also fast”__
> >
> > In Figure 11 of the manuscript, we ablate M, and find that our method is not sensitive to this hyperparameter for values $M \in \{ 5, 10, 20 \}$. Smaller values of M perform marginally worse than larger values of $M$, and have the advantage of a smaller computational cost. Based on the cost analysis, DA-Fusion is only 20.2% more expensive than RandAugment according to wall-clock time in our configuration, and performs 24.2% better for fine-grain concepts.

---

> ### Author Response · Authors · 2023-11-21
> **Following Up On The Rebuttal**
>
> Thank you for reviewing our manuscript, we would like to follow up on our rebuttal.  If there are any remaining points that you would like us to address, please let us know.  Otherwise, thank you for your review and we look forward to your response.

---

> > ### Comment · Reviewer_KxYv · 2023-11-21
> >
> > Thanks for the additional analysis and discussion!
> >
> > One clarification, DA-fusion is said to have "no prior knowledge about the image content", but for Control Method 2 (the case where there are co-occuring objects) DA-fusion seems to requires knowledge about the image content to distinguish the cat and dog. Did I understand this correctly? For the datasets that are less object-centric (with co-occuring objects) is the prompt used to perform textual inversion still "a photo of $w_i$" or does it depend on the class/image etc? Thanks!

---

> ### Author Response · Authors · 2023-11-21
> **Additional Clarifications**
>
> Thank you for your questions and discussion!
>
> __"for Control Method 2 (the case where there are co-occuring objects) DA-fusion seems to requires knowledge about the image content to distinguish the cat and dog. Did I understand this correctly?"__
>
> If __Control Method 2__ is used, it benefits from a prompt that describes which object to focus on, but note this is an *additional way to control DA-Fusion* and not a core part of the method. DA-Fusion does not require knowledge about the image content to generate performant augmentations. In Figure 5-11 of the manuscript, we do not provide image-level knowledge to DA-Fusion, and we see strong performance improvements between 12% and 24% depending on how fine-grain or rare concepts are.
>
> __"For the datasets that are less object-centric (with co-occuring objects) is the prompt used to perform textual inversion still "a photo of $w_i$ " or does it depend on the class/image etc?"__
>
> The prompt format in textual inversion is the same for all datasets and does not depend on the class/image. We use the prompt format "a photo of a <class>" where "<class>" is a learned pseudo-prompt. This choice of prompt (part of enforcing uniform hyperparameters across all datasets) ensures that DA-Fusion is not fine-tuned to a particular kind of dataset.
>
> Thank you for the discussion, please let us know if you have additional questions. We look forward to your response.

---

> > ### Comment · Reviewer_KxYv · 2023-11-23
> >
> > Thanks for the clarification. My main concern would then still be on the generalizability of the method, that e.g., Control Method 2 requires image level information to learn the correct object. Thus, I am inclined to keep my score.

---

> ### Author Response · Authors · 2023-11-23
> **Clarifications For Generalizability**
>
> Thanks for following up on the generalizability of the method, we appreciate your clarification to your earlier concern. In this response, we will show that DA-Fusion does not use image level information to learn the correct object. We appreciate the back-and-forth discussion you had with us during the peer review process, and hope our new points on generalizability can help resolve your concern.
>
> # Learning The Correct Object Without Image-Level Information
>
> __“My main concern would then still be on the generalizability of the method, that e.g., Control Method 2 requires image level information to learn the correct object.”__
>
> DA-Fusion infers which object is the correct one purely from shared statistics in the images, and does not use image-level information to identify the correct object. __Control Method 2__ is *an additional mechanism* that we don’t use in our evaluation in Figures 5-11, but can be used by practitioners who want additional hands-on control over what DA-Fusion generates. The primary way to instruct DA-Fusion what to generate is by choosing example images for learning word embeddings that contain the desired visual concept. We show this procedure identifying the correct object when the example images have co-occurring objects, and partial occlusions.
>
> Examples of images from COCO with co-occurring objects, and partial occlusions shown to DA-Fusion are uploaded at the following anonymous link. In this example, we show DA-Fusion partially occluded cats in different poses with co-occurring objects, like the stuffed elephant blocking the cat in example image four. DA-Fusion correctly focuses on the cats, shown by the generations uploaded at the anonymous link, despite co-occurring objects, and occlusions.
> https://drive.google.com/drive/folders/1wSc4D-XXXJ-AzjO6SbOFy_YBng9YD64F?usp=sharing
>
> ---
>
> In summary, DA-Fusion does not use image level information (e.g no image-specific prompt) to learn the correct object, and only requires example images that contain the desired concept. The generations at the link above show DA-Fusion is robust to poorly-chosen examples with co-occurring objects, and occlusions.

---

> > ### Comment · Reviewer_KxYv · 2023-11-23
> >
> > Thanks again for the clarification, I misunderstood the purpose of the control methods. I will raise my score from 5 to 6. Also given that there are many follow up works on textual inversion that e.g., amortizes the optimization, or extracts several fine-grained concepts etc., which may be orthorgonal to the method, but it would be interesting to see how much of an improvement it makes.

---

### Official Review · Reviewer_rXxt · 2023-10-31

**Soundness:** 3 good
**Presentation:** 3 good
**Contribution:** 3 good
**Rating:** 8
**Confidence:** 4

**Summary:**

The paper introduces a novel approach to data augmentation in deep learning, addressing the issue of limited diversity in traditional augmentation methods. The authors propose a method called DA-Fusion, which leverages large pretrained generative models to generate variations of real images while respecting their semantic attributes. They fine-tune pseudo-prompts to instruct the diffusion model on what to augment, and their approach is tested on few-shot image classification tasks across various domains, including a real-world weed recognition task.

The authors provide a comprehensive review of related work, highlighting the advantages of diffusion models in image generation and editing. They also discuss the challenges of preventing leakage of internet data when using large pretrained generative models for synthetic data generation.

The experimental results show that DA-Fusion consistently improves accuracy in various domains, outperforming traditional data augmentation methods. The paper includes a detailed analysis of the results, including a breakdown by the presence of common, fine-grain, or completely new concepts in the datasets.

The paper concludes by suggesting future directions for improving the flexibility and performance of their method, such as better control over image augmentation, maintaining temporal consistency in decision-making settings, and enhancing the photo-realism of the diffusion model backbone.

Overall, the paper presents a promising approach to data augmentation that addresses the limitations of traditional methods and demonstrates its effectiveness in various domains. The discussion of potential future improvements adds value to the work.

**Strengths:**

The work on DA-Fusion offers several strengths and introduces significant novelty in the context of data augmentation:
* Novel Data Augmentation Technique: DA-Fusion introduces a unique approach to data augmentation by leveraging large pretrained generative models. It goes beyond traditional data augmentation methods that mainly involve geometric transformations. This novelty lies in utilizing generative models to create diverse and semantically meaningful variations of real images.
* Semantic Preservation: Unlike traditional data augmentation techniques that focus on basic geometric transformations, DA-Fusion aims to respect the semantic attributes of images. It modifies images in a manner that retains their underlying semantics, such as the design of objects or specific visual details, making it more relevant for real-world recognition tasks.
* Few-shot Learning Improvement: The paper demonstrates the effectiveness of DA-Fusion in enhancing few-shot image classification tasks. It is particularly valuable in scenarios where only a limited number of real images per class are available. This is a significant advantage as few-shot learning is a challenging problem with practical applications.
* Fine-Grained Concepts: The authors show that DA-Fusion is especially beneficial for fine-grained concepts, where traditional data augmentation methods may not provide sufficient diversity. This highlights the method's potential in improving the recognition of subtle visual differences in image classification tasks.
* Leakage Mitigation: The paper addresses the important issue of preventing leakage of internet data, which is often a concern when utilizing large pretrained generative models. The proposed defenses to mitigate leakage during evaluation contribute to the robustness and reliability of the method.
* Generalization to New Concepts: DA-Fusion's ability to generalize to new concepts not seen during the diffusion model's training is a noteworthy aspect. This is a crucial capability, as it allows the method to adapt to a wide range of recognition tasks without the need for extensive additional data.
* Clear Experimental Validation: The work provides a thorough experimental validation, including results on a variety of datasets spanning common, fine-grained, and completely new concepts. The consistent improvement in accuracy across different domains demonstrates the practical applicability and versatility of the approach.

**Weaknesses:**

Here are some of the weaknesses of this work:
* Complexity and Computational Cost: The proposed method involves fine-tuning pseudo-prompts for each concept, which can be computationally expensive and time-consuming. This approach might not be as practical as traditional data augmentation techniques that are computationally efficient and easy to implement.
* Lack of Control Over Augmentations: While DA-Fusion introduces the concept of modifying images while respecting their semantic attributes, it does not explicitly provide fine-grained control over how the augmentations are performed. This lack of control may limit its applicability to specific use cases where precise image modifications are required.

Some nitpicks: Citations starting from second page are not in brackets like that in the first page. Would be nice to have consistency and keep them in brackets.

**Questions:**

I believe the idea is nice and clear. Paper is well written. Some might be of the opinion that the idea is not novel. However, I do like the effective use of pre-trained diffusion models and textual inversion for data augmentation. I like the detailed analysis that the authors have done for their approach.

One thing I would like to see is how this method can be used to generate positive and negative samples for contrastive learning based approaches. I understand this can be a herculean task for the authors to do now so not required as of now. But curious to know about the effect of this data aug strategy in SSL.

---

> ### Author Response · Authors · 2023-11-20
> **Response To Reviewer rXxt (1/2)**
>
> Thank you for reviewing our paper and for your valuable feedback! We appreciate the thorough review, your enthusiastic points, and positive impression of the manuscript. We hope to answer your questions and discuss the weaknesses you listed in our rebuttal. The main points listed by this reviewer center around the complexity and computational cost of DA-Fusion, controlling the augmentations generated by DA-Fusion, and extensions to self-supervised learning tasks.
>
> We address these points, organized by category, in the following sections.
>
> # Complexity and Computational Cost
>
> __“This approach might not be as practical as traditional data augmentation techniques that are computationally efficient and easy to implement.”__
>
> The trade-off of performance and computational cost is important for users considering the viability of generative data augmentations. We agree the cost of DA-Fusion is higher than traditional data augmentation, but our method is parallelizable, and the cost is once upfront, whereas the cost of traditional data augmentation is at every training iteration.
>
> We provide statistics for training a classifier with DA-Fusion in the following table, assuming a dataset with 100 classes, 4 images per class, a training batch size of 32, trained for 10,000 steps, with M=5. The machine has 4 Nvidia RTX 6000 ada GPUs, and image generation is parallelized across all the GPUs.
>
> | Method | Total Time (minutes) |
> | ----------- | ----------- |
> | DA-Fusion (25-step diffusion sampler) | 21:19 |
> | DA-Fusion (12-step diffusion sampler) | 14:47 |
> | RandAugment | 12:18 |
> | No Augmentation | 08:15 |
>
> Training a classifier with DA-Fusion is only 20.2% more expensive than RandAugment with the 12-step diffusion sampler according to wall-clock time in this configuration, and performs up to 24.2% better for fine-grain concepts, which is discussed in the paper in Section 6, Figure 6 in the manuscript. We are adding this cost analysis to the paper.
>
> __“The proposed method involves fine-tuning pseudo-prompts for each concept, which can be computationally expensive and time-consuming.”__
>
> We agree that fine-tuning pseudo-prompts is time-consuming. Each concept requires 10 minutes of fine-tuning on our machine, and this can be parallelized. One fine-tuning experiment requires 11GB of RAM, and we are able to fit 4 of these in parallel on an NVIDIA RTX 6000 ada GPU, resulting in 16 parallel fine-tuning experiments with 4 GPUs. With our 100 class dataset, this requires 10 * 100 / 16 = 62.5 minutes of time spent fine-tuning.
>
> Faster methods for learning pseudo-prompts are being developed. Recent work by Voronov et al. 2023 (https://arxiv.org/abs/2302.04841) could reduce this cost by a factor of 8, resulting in a total cost of 7.8 minutes of time spent fine-tuning. Additionally, HyperNetworks are being developed (Ruiz et al. 2023 https://arxiv.org/abs/2307.06949) that may eliminate the fine-tuning step by directly predicting the pseudo-prompt embeddings with a negligible cost.

---

> > ### Comment · Reviewer_rXxt · 2023-11-22
> >
> > Thanks for the response. The authors addressed my concerns and I would like to keep my score.

---

> ### Author Response · Authors · 2023-11-20
> **Response To Reviewer rXxt (2/2)**
>
> # Controlling The Augmentation
>
> __“This lack of [fine-grained control over how the augmentations are performed] may limit its applicability to specific use cases where precise image modifications are required.”__
>
> Generative models are uniquely promising for data augmentation because they allow the user to control the content and style of images in training datasets. DA-Fusion implements three simple interfaces that control the content and style of what is generated by our method.
>
> 1) guiding what’s generated with prompts
> 2) guiding what Textual Inversion learns with prompts
> 3) erasing the ability to generate specific concepts written in prompts
>
> Each control method has a simple natural language interface.
>
> __Control Method 1__: The word embeddings learned by DA-Fusion are flexible, and the user can modify the content and style of generations after learning the word embeddings. For example, if the prompt “a photo of a <class>” leads to the images named “original generation” at the following anonymous link, changing the prompt to “a photo of a <class> wearing a bow-tie”, leads to the images named “generation with bow-tie” at the following anonymous link:
> https://drive.google.com/drive/folders/1gnCa0SBd9tadL9o2SK0jmNNYoNvC76Sc?usp=sharing
> We are updating the manuscript to discuss __Control Method 1__ in the appendix.
>
> __Control Method 2__: Suppose we show DA-Fusion the image named “original image” at the following anonymous link and learn a “<class>” word embedding using this image. The image has a cat and a dog. If we want the “<class>” word embedding to focus on the cat and ignore the dog, we can prompt Stable Diffusion with “a photo of a <class> sitting on the right of a dog” when learning the “<class>” word embedding. If we generate images using an inference-time prompt “a photo of a <class>” the generations only have cats, showing the “<class>” word embedding has ignored the dog and focused on the cat in the original image.
> https://drive.google.com/drive/folders/1lmr7e7qBZHwyGJyIkxt-0bUAxcn0934M?usp=sharing
> We are updating the manuscript to discuss __Control Method 2__ in the appendix.
>
> Controlling the style of what tokens learn can be done with Textual Inversion. We build on the Textual Inversion script released by HuggingFace, and by using `--learnable_property=style` the user can switch modes and cause tokens to learn style rather than object-based concepts.
>
> __Control Method 3__: Finally, we can erase Stable Diffusion’s generation ability for specific concepts. Suppose we generate images of roads using the prompt “a photo of a <class>” and we realize that cars often show up in the generations, but our real data never has cars. We can erase the “car” concept and prevent Stable Diffusion from generating it, resulting in the following images. Additional examples for erasing concepts are in Figure 7 of the manuscript.
> https://drive.google.com/drive/folders/1eDkXFKhpN94ZEhQtih97PdXsJtjzqKuz?usp=sharing
> We discuss __Control Method 3__ in Section 6.1 of the manuscript.
>
> # Questions & Miscellaneous
>
> __“One thing I would like to see is how this method can be used to generate positive and negative samples for contrastive learning based approaches.”__
>
> We think this is a promising next direction as well, and agree with the reviewer that it would require a significant amount of work to include a proper empirical analysis. The inclusion is beyond the scope of this paper, but we are happy to discuss what this extension would entail.
>
> Generative data augmentations like DA-Fusion can edit images to interpolate between different classes. With this in mind, to generate highly informative positive and negative augmentations that “straddle” the decision boundary of two classes, we can start from images of class A, and transform them into images of class B that resemble class A. These samples are positive for class B, and negative for class A. This may be helpful for classes that are often confused.
>
> __“Some nitpicks: Citations starting from second page are not in brackets like that in the first page. Would be nice to have consistency and keep them in brackets.”__
>
> Thanks for catching this! We are updating the citation format in the manuscript to keep citations consistent by changing citations to brackets from the second page and after.

---

### Meta-Review · Area_Chair_p5Z2 · 2023-12-11

**Metareview:**

This paper introduces DA-Fusion, a data augmentation approach that leverages pretrained generative models to generate variations of real image while respecting their semantic attributes. The authors present experimental results demonstrating the effectiveness of DA-Fusion across various domains.

The paper received positive feedback with ratings of 8, 8, 6, 6.
  The reviewers appreciated the contribution of the paper and acknowledge the effectiveness of DA-Fusion shown by extensive experiments in diverse contexts.
  Initial concerns were raised by the reviewers regarding the complexity, computational cost, and generalizability of the method. The authors effectively addressed these concerns in their rebuttal, offering clarifications and updates to the manuscript.

Consequently, all reviewers reached a consensus, recommending the acceptance of the paper. Congratulations to the authors on their nice work!

**Justification For Why Not Higher Score:**

I'm on the fence about whether to raise the score of this paper to Accept (Spotlight). While I believe the quality of the paper is pretty high, there are a few aspects that make me hesitate to give it an even higher score. My major concern is about the execution of the paper, such as the discussion about generality, additional computational cost, and comparisons with more advanced data augmentation baselines.

**Justification For Why Not Lower Score:**

It is a paper with clear acceptance based on the consensus reached by reviewers.

---

### Decision · Program_Chairs · 2024-01-16

Accept (poster)